# Effects of Superabsorbent Polymers on Growth and Pigment Allocation in *Chlorella vulgaris*

**DOI:** 10.3390/plants14192962

**Published:** 2025-09-24

**Authors:** Gabriella Erzsébet Szemők, László Balázs, Ákos Tarnawa, Szandra Klátyik, Gergő Péter Kovács, Zoltán Kende

**Affiliations:** 1Institute of Agronomy, Hungarian University of Agriculture and Life Sciences, H-2100 Gödöllő, Hungary; 2Agro-Environmental Research Centre, Institute of Environmental Sciences, Hungarian University of Agriculture and Life Sciences, H-2100 Gödöllő, Hungary

**Keywords:** superabsorbent polymers (SAPs), *Chlorella vulgaris*, pigment allocation, drought stress mitigation, algae–microbe interactions, sustainable crop production

## Abstract

Superabsorbent polymers (SAPs) are increasingly applied in agriculture to enhance soil water retention, reduce nutrient loss, and mitigate drought stress—challenges expected to intensify under global climate change. While their benefits for crop growth are well documented, much less is known about their influence on free-living microorganisms. Here, we examined the effects of three SAP chemistries—potassium polyacrylate (DCM Aquaperla^®^), starch-based polyacrylamide (Zeba Plus SP^®^), and γ-polyglutamate (Stockosorb^®^ 660 Medium)—on the growth and pigment composition of Chlorella vulgaris Beijerinck across three initial cell densities (22.8 × 10^3^, 228 × 10^3^, and 2.228 × 10^6^ cells/mL). Six spectral indices, derived from weekly absorbance measurements over seven weeks, were used to track biomass and pigment allocation. Nonparametric repeated-measures analysis and principal component analysis revealed strong effects of SAP type, algal density, and time. Zeba consistently maintained biomass comparable to the control while enhancing carotenoid- and xanthophyll-sensitive indices, suggesting pigment reallocation without growth suppression. Stockosorb produced intermediate responses, whereas Aquaperla frequently reduced biomass-related measures, particularly at high density. Pigment allocation was also density-dependent, with low-density cultures investing proportionally more in carotenoids. Overall, these results show that SAP–microbe interactions are strongly influenced by polymer chemistry and starting biomass, with implications for biotechnology, environmental risk assessment, and sustainable crop production systems that aim to support both algal and plant resilience under drought.

## 1. Introduction

Soil water scarcity and inefficient water use remain major constraints on sustainable crop production, particularly in arid and semi-arid regions [1]. Superabsorbent polymers (SAPs), especially hydrogels such as potassium polyacrylate, are increasingly applied as soil amendments to enhance water retention [2], reduce nutrient leaching, and improve plant performance under drought conditions [3,4]. Biodegradable and hybrid SAPs combining synthetic polymers with starch or chitosan balance high water-holding capacity with environmental sustainability [5,6] and can be tailored for controlled fertilizer release [7,8]. Chitosan-based and keratin–geopolymer hydrogels derived from agricultural waste offer additional eco-friendly options [9,10]. Long-term field studies show that the SAP application can increase soil aggregate stability, raise organic carbon and microbial biomass, and significantly enhance crop yield and water-use efficiency [11,12,13]. In sandy loam soils, SAPs combined with biofertilizers have improved nutrient status and organic matter, illustrating their role in integrated approaches to sustainable crop production [9,14].

While the benefits of SAPs in soil–plant systems are well-documented, their compatibility with microorganisms is less explored. In biologically active environments such as soils, hydroponic systems, or algal cultures, it is important to determine whether SAPs inhibit or support microbial viability [15]. Microalgae, particularly *Chlorella vulgaris* Beijerinck, are versatile biotechnological resources [16] with applications in bioenergy, wastewater remediation [17], aquaculture feed [18,19,20], and the production of high-value metabolites such as pigments and polyunsaturated fatty acids [15,21,22]. Their high growth rates, photosynthetic efficiency, and capacity to accumulate energy-rich compounds make them ideal for sustainable production systems [23,24]. Cultivation in photobioreactors, however, is constrained by water management, nutrient stability, and maintenance of optimal densities [23,25].

For successful integration of *C. vulgaris* into agro-biotechnological systems, it is essential to test whether amendments that improve water and nutrient dynamics—such as SAPs—are compatible with living microbial communities and do not cause adverse effects. Existing work on SAP–microbe interactions focuses mainly on hydrogel-immobilized microalgae for wastewater treatment or bioactive compound production [13,26,27,28,29]. Immobilization studies show that biocompatible matrices can preserve viability, sustain metabolism, and enhance stress tolerance. Hydrogel–microbe systems can also be tuned for nutrient release, pollutant removal, and microenvironment control [30,31,32]. However, a mechanistic understanding of the non-immobilizing effects of SAP on algal growth, pigment composition, and spectral properties is lacking.

We selected these three SAPs to represent distinct chemical classes and modes of action that are relevant to agricultural use. Aquaperla (Potassium polyacrylate) is a widely used synthetic hydrogel known for its strong cation exchange capacity and high water retention, but it may cause ionic imbalances. Zeba is starch-based and biodegradable, representing renewable biopolymers that can release nutrients during degradation. Stockosorb (γ-polyglutamate) is a microbial-derived biopolymer with high water-holding capacity and excellent biocompatibility. By comparing these contrasting SAP chemistries, we aimed to test whether algal growth responses depend on polymer origin (synthetic vs. biodegradable vs. microbial-derived) and ionic properties, providing a first step toward assessing SAP–microbe compatibility in agro-ecosystems.

This study investigates the effects of three SAP chemistries—potassium polyacrylate (Aquaperla), starch-based polyacrylamide-co-acrylic acid potassium salt (Zeba), and γ-polyglutamate (Stockosorb)—on the growth and pigment composition of *C. vulgaris* across varying starting densities. Potassium polyacrylate offers exceptional water retention and moisture release, improving plant biomass in terrestrial systems [11,12,13]. Starch-based hydrogels show high swelling capacity, biodegradability, and nutrient regulation, aiding crop establishment under low-water conditions [6,9,14]. γ-polyglutamate is likewise biodegradable, water-retentive, and biocompatible, making it a promising candidate for supporting living cultures [7,33].

Changes in pigment allocation are critical because they reflect shifts between growth-oriented chlorophyll synthesis and stress-related carotenoid/xanthophyll accumulation [34]. Such reallocations not only determine algal resilience to light and osmotic stress [35,36,37] but also influence the economic potential of cultures, as carotenoids are high-value metabolites [38,39].

This work tests the hypothesis that SAPs, when introduced into living microbial systems, will not cause complete growth inhibition but may alter biomass accumulation and pigment allocation depending on polymer chemistry and algal density. The findings will provide insight into SAP–microbe compatibility, supporting their safe and effective use in sustainable agriculture, soil amendment practices, and integrated biotechnological systems [27,40].

## 2. Results

### 2.1. Cross-Sectional Comparisons of SAP Treatments on Algal Growth

#### 2.1.1. Integrated Absorbance

Within each algal density, Integrated Absorbance differed among SAP treatments (22.8 × 10^3^: H = 13.50, df = 3, *p* = 0.0037; 228 × 10^3^: H = 12.64, df = 3, *p* = 0.0055; 2.228 × 10^6^: H = 13.79, df = 3, *p* = 0.0032), indicating that hydrogel composition modulated bulk absorbance at all densities. Dunn–Bonferroni post hoc tests revealed selective, density-dependent contrasts rather than a uniform ranking (Figure 1). At the lowest density (22.8 × 10^3^), Control exceeded Stockosorb (*p*.adj = 0.0038), while Control did not differ from Zeba and Aquaperla (letters “a” for Control, “ab” for Aquaperla/Zeba, “b” for Stockosorb). At the mid-density (228 × 10^3^), Control exceeded Stockosorb again (*p*.adj = 0.0065); Zeba remained statistically indistinguishable from Control, and Aquaperla did not differ from either Control or Stockosorb (letters “a” for Control, “ab” for Aquaperla/Zeba, “b” for Stockosorb). At the highest density (2.228 × 10^6^), Control exceeded Aquaperla (*p*.adj = 0.0022) but did not differ from Stockosorb or Zeba (letters “a” for Control, “b” for Aquaperla, “ab” for Stockosorb/Zeba). In sum, Control showed targeted advantages over Stockosorb (low and mid-densities) and over Aquaperla (high density), whereas Zeba consistently matched Control across densities, supporting the view that algae maintained control-like absorbance levels in Zeba hydrogels. These rank-based patterns align with the raw-scale descriptives: at 22.8 × 10^3^, Control (65.5 ± 6.9; 59.7–75.0) exceeded Stockosorb (30.2 ± 5.5; 24.2–36.5) and Aquaperla (36.5 ± 5.6; 28.3–40.0), with Zeba intermediate (44.9 ± 1.5; 43.3–46.5); at 228 × 10^3^, Control (89.7 ± 3.3; 85.0–92.4) again led Stockosorb (46.9 ± 6.4; 38.8–53.3) and Aquaperla (55.8 ± 10.7; 47.7–71.6), with Zeba close to Control (76.8 ± 7.3; 66.4–83.0); and at 2.228 × 10^6^, Control remained highest (169.6 ± 21.5; 151.5–194.5), followed by Zeba (111.8 ± 3.7; 106.3–114.5), Stockosorb (101.6 ± 6.7; 95.3–110.9), and Aquaperla (82.2 ± 7.2; 72.5–89.3).

#### 2.1.2. Chlorophyll Index

Within each algal density, Chlorophyll Index differed among SAP treatments (22.8 × 10^3^: H = 13.26, df = 3, *p* = 0.0041; 228 × 10^3^: H = 11.93, df = 3, *p* = 0.0076; 2.228 × 10^6^: H = 12.90, df = 3, *p* = 0.0048), indicating density-specific modulation of chlorophyll-related absorption by the hydrogels. Dunn–Bonferroni post hoc tests showed that the differences were selective rather than global (Figure 2). At the lowest density (22.8 × 10^3^), Control exceeded Zeba (*p*.adj = 0.0050), whereas Aquaperla and Stockosorb were statistically indistinguishable from both Control and Zeba after adjustment (plot letters “a” for Control, “ab” for Aquaperla/Stockosorb, “b” for Zeba). At the mid-density (228 × 10^3^), only Control exceeded Stockosorb (*p*.adj = 0.0065); contrasts involving Aquaperla or Zeba did not survive Bonferroni correction despite visible spacing in the boxplots (letters “a” for Control, “ab” for Aquaperla/Zeba, “b” for Stockosorb). At the highest density (2.228 × 10^6^), Control exceeded Aquaperla (*p*.adj = 0.0022), while Control vs. Stockosorb and Control vs. Zeba were not significant after adjustment (letters “a” for Control, “b” for Aquaperla, “ab” for Stockosorb/Zeba). Thus, Control shows targeted advantages—against Zeba at low density, against Stockosorb at mid density, and against Aquaperla at high density—while Zeba is often statistically similar to Control, consistent with preserved chlorophyll signal in that gel. These rank-based findings align with the raw-scale descriptives: at 22.8 × 10^3^, Control (0.285 ± 0.037; 0.254–0.335) exceeded Aquaperla (0.107 ± 0.008; 0.100–0.118), Stockosorb (0.081 ± 0.009; 0.073–0.094), and Zeba (0.073 ± 0.004; 0.068–0.078); at 228 × 10^3^, Control (0.399 ± 0.016; 0.376–0.411) remained highest relative to Aquaperla (0.175 ± 0.017; 0.158–0.194), Stockosorb (0.151 ± 0.025; 0.125–0.185), and Zeba (0.204 ± 0.016; 0.180–0.217); and at 2.228 × 10^6^, Control (0.746 ± 0.103; 0.658–0.861) exceeded Aquaperla (0.295 ± 0.023; 0.264–0.319), with Stockosorb (0.380 ± 0.028; 0.350–0.417) and Zeba (0.361 ± 0.018; 0.334–0.373) intermediate and overlapping in the post hoc tests.

#### 2.1.3. Blue/Red Ratio

Kruskal–Wallis tests indicated among-SAP differences in the Blue/Red Ratio (BR) at all algal densities (22.8 × 10^3^: H = 8.89, df = 3, *p* = 0.0308; 228 × 10^3^: H = 12.73, df = 3, *p* = 0.0053; 2.228 × 10^6^: H = 14.12, df = 3, *p* = 0.0027), showing that hydrogel composition modulated relative blue vs. red absorbance. Dunn–Bonferroni post hoc comparisons revealed (Figure 3) that the pattern was driven primarily by Zeba, with density-specific advantages: at 22.8 × 10^3^, Zeba > Stockosorb (*p*.adj = 0.036), whereas differences between Zeba and Control or Aquaperla did not survive correction (letters typically “a” for Zeba, “ab” for Control/Aquaperla, “b” for Stockosorb). At 228 × 10^3^, Zeba > Aquaperla (*p*.adj = 0.0022); Zeba vs. Control and Zeba vs. Stockosorb were not significant after correction (letters “a” for Zeba, “b” for Aquaperla, “ab” for Control/Stockosorb). At 2.228 × 10^6^, Zeba > Control (*p*.adj = 0.0022), while contrasts involving Aquaperla or Stockosorb were not significant (letters “a” for Zeba, “b” for Control, “ab” for Aquaperla/Stockosorb). These results indicate that Zeba frequently produced higher BR values than at least one comparator within each density, consistent with gel-specific shifts in pigment optics rather than a generalized attenuation of spectral signal. Descriptives in the raw scale mirrored the rank tests: at 22.8 × 10^3^, Zeba showed clearly elevated BR (1.585 ± 0.088; 1.476–1.665) relative to Control (1.095 ± 0.026; 1.063–1.127), Aquaperla (1.107 ± 0.154; 0.959–1.297), and Stockosorb (1.042 ± 0.133; 0.879–1.191); at 228 × 10^3^, Zeba remained highest (1.342 ± 0.046; 1.304–1.406) compared with Control (1.083 ± 0.009; 1.070–1.093), Aquaperla (1.036 ± 0.048; 0.966–1.069), and Stockosorb (1.091 ± 0.026; 1.073–1.127); and at 2.228 × 10^6^, Zeba again led (1.176 ± 0.021; 1.146–1.191) over Control (0.976 ± 0.043; 0.931–1.015), Aquaperla (1.071 ± 0.006; 1.064–1.076), and Stockosorb (1.046 ± 0.007; 1.038–1.056). Although these mean differences are visually pronounced, only the specific pairs above met the conservative Bonferroni threshold.

#### 2.1.4. Normalized Difference Algal Index (NDAI)

Kruskal–Wallis tests showed among-SAP differences in NDAI at the lowest and highest algal densities (22.8 × 10^3^: H = 11.80, df = 3, *p* = 0.0081; 2.228 × 10^6^: H = 12.73, df = 3, *p* = 0.0053), but not at the mid-density (228 × 10^3^: H = 6.68, *p* = 0.083). Dunn–Bonferroni post hoc tests indicated that the significant omnibus results were driven by Zeba vs. Control: at 22.8 × 10^3^, Zeba > Control (*p*.adj = 0.0038) and at 2.228 × 10^6^, Zeba > Control again (*p*.adj = 0.0022). All other pairs at these densities and all pairs at 228 × 10^3^ did not remain significant after Bonferroni correction (Figure 4). Because NDAI values are negative in our dataset, “higher NDAI” means less negative (numerically closer to zero), which we interpret as a relatively greater algal signal on this index. The descriptive statistics mirror the rank findings. At 22.8 × 10^3^, Zeba showed the highest (least negative) NDAI (−0.301 ± 0.016; range −0.321 to −0.283), followed by Stockosorb (−0.383 ± 0.059; −0.448 to −0.314) and Aquaperla (−0.412 ± 0.104; −0.554 to −0.316), while Control was lowest (−0.614 ± 0.023; −0.642 to −0.594). At 228 × 10^3^, Zeba (−0.497 ± 0.034; −0.530 to −0.450), Stockosorb (−0.542 ± 0.053; −0.573 to −0.462), and Aquaperla (−0.578 ± 0.188; −0.855 to −0.442) appeared less negative than Control (−0.734 ± 0.006; −0.742 to −0.727), but the omnibus test did not reach significance, indicating that dispersion and sample size likely limited power at this density. At 2.228 × 10^6^, Zeba again had the highest (least negative) NDAI (−0.655 ± 0.016; −0.665 to −0.632), exceeding Control (−0.787 ± 0.004; −0.791 to −0.783) in the post hoc test; Aquaperla (−0.711 ± 0.034; −0.751 to −0.672) and Stockosorb (−0.715 ± 0.008; −0.724 to −0.704) were intermediate but not significantly different from Control after adjustment. Overall, NDAI supports a density-dependent advantage for Zeba (vs. Control) at the lowest and highest starting biomasses, with no confirmed separations at the mid-density.

#### 2.1.5. Photochemical Reflectance Index (PRI)

Kruskal–Wallis tests indicated that PRI differed among SAP treatments only at the mid-algal density (228 × 10^3^: H = 12.90, df = 3, *p* = 0.0048), whereas omnibus tests at the lowest (22.8 × 10^3^: H = 7.13, *p* = 0.068) and highest (2.228 × 10^6^: H = 7.17, *p* = 0.0667) densities were not significant. Post hoc Dunn–Bonferroni comparisons (Figure 5) at 228 × 10^3^ showed that Zeba exceeded both Aquaperla (*p*.adj = 0.0360) and Stockosorb (*p*.adj = 0.0084); contrasts involving Control did not survive adjustment, placing Control statistically between Zeba and the other gels (plot letters: “b” for Zeba; “a” for Control/Aquaperla/Stockosorb). At 22.8 × 10^3^ and 2.228 × 10^6^, no pairwise differences remained after Bonferroni correction (letters largely shared across SAPs), indicating that PRI separation is density-dependent and most pronounced at intermediate biomass. Descriptives on the original scale align with these rank findings. At 22.8 × 10^3^, Zeba exhibited the most positive PRI (0.0387 ± 0.0074; 0.0308–0.0467), while Control (−0.0091 ± 0.0105; −0.0205–0.0030), Aquaperla (0.0080 ± 0.0180; −0.0143–0.0283), and Stockosorb (−0.0013 ± 0.0362; −0.0285–0.0483) clustered near zero with wide overlap. At 228 × 10^3^, Zeba was clearly higher (0.0300 ± 0.0051; 0.0238–0.0359) than Aquaperla (−0.00023 ± 0.00755; −0.0114–0.00444) and Stockosorb (0.00114 ± 0.00187; −0.00061–0.00360), with Control intermediate (0.00895 ± 0.00101; 0.00772–0.01019). At 2.228 × 10^6^, all SAPs showed small positive PRI values with modest spread—Control 0.0184 ± 0.0027 (0.0155–0.0211), Aquaperla 0.0196 ± 0.0039 (0.0156–0.0247), Stockosorb 0.0171 ± 0.0018 (0.0157–0.0196), and Zeba 0.0237 ± 0.0027 (0.0206–0.0261)—but these differences did not reach significance after multiplicity control. Overall, PRI indicates selective enhancement under Zeba at the intermediate density, with broadly comparable photochemical reflectance responses across SAPs at low and high densities.

#### 2.1.6. Normalized Phaeophytinization Index

NPQI differed among SAP treatments at all algal densities (22.8 × 10^3^: H = 8.82, df = 3, *p* = 0.0317; 228 × 10^3^: H = 8.32, df = 3, *p* = 0.0399; 2.228 × 10^6^: H = 11.27, df = 3, *p* = 0.0103), indicating hydrogel-dependent shifts in pigment state. Dunn–Bonferroni tests showed (Figure 6) a clear, density-robust pattern: Zeba exceeded Control at every density (22.8 × 10^3^: *p*.adj = 0.0451; 228 × 10^3^: *p*.adj = 0.0360; 2.228 × 10^6^: *p*.adj = 0.0286), and at the highest density Aquaperla also exceeded Control (*p*.adj = 0.0227). All other pairwise contrasts did not remain significant after Bonferroni correction, despite some visible spacing in the boxplots—particularly at the lowest density where Stockosorb’s mean was high, but dispersion was large. Because higher NPQI values (i.e., less negative or more positive) are typically interpreted as greater pheophytinization, these results suggest that algae survived in gels while exhibiting modest, gel-specific pigment adjustments, rather than wholesale loss of optical signal. Descriptives on the original scale align with the rank pattern. At 22.8 × 10^3^, Zeba (−0.018 ± 0.001; range −0.020 to −0.017) and Aquaperla (−0.002 ± 0.078; −0.068 to 0.110) were less negative than Control (−0.110 ± 0.015; −0.123 to −0.088), with Stockosorb showing the largest spread (0.046 ± 0.176; −0.080 to 0.306). At 228 × 10^3^, Zeba again led (−0.057 ± 0.002; −0.059 to −0.054) relative to Control (−0.126 ± 0.005; −0.131 to −0.119), with Stockosorb (−0.072 ± 0.002; −0.074 to −0.070) and Aquaperla (−0.082 ± 0.080; −0.198 to −0.013) intermediate. At 2.228 × 10^6^, Zeba (−0.093 ± 0.004; −0.095 to −0.087) and Aquaperla (−0.092 ± 0.006; −0.100 to −0.086) exceeded Control (−0.135 ± 0.003; −0.137 to −0.130), with Stockosorb slightly higher than Control but not significant after adjustment (−0.099 ± 0.004; −0.105 to −0.096). Overall, NPQI reveals consistent, statistically confirmed elevations for Zeba versus Control across densities, and an additional Aquaperla advantage at the highest density, reinforcing the interpretation that hydrogels support algal persistence with density-dependent pigment state shifts.

### 2.2. Longitudinal Nonparametric Analysis

#### 2.2.1. Integrated Absorbance

The nonparametric repeated-measures model indicated (Table 1) strong main effects of SAP treatment (ATS = 91.38, df = 2.34, *p* < 0.001), algae density (ATS = 456.76, df = 1.81, *p* < 0.001), and time (ATS = 761.83, df = 2.95, *p* < 0.001), together with all two-way and three-way interactions (SAP × Algae: ATS = 5.72, df = 4.08, *p* < 0.001; SAP × Time: ATS = 20.22, df = 5.55, *p* < 0.001; Algae × Time: ATS = 41.90, df = 5.13, *p* < 0.001; SAP × Algae × Time: ATS = 12.95, df = 8.26, *p* < 0.001). These results show that (i) integrated absorbance rose over weeks in a nonparametric sense, (ii) higher starting cell densities generally achieved higher levels, and (iii) the shape and height of the time courses depended on both SAP and density.

Figure 7 displays Relative Treatment Effects (RTEs; probability of exceeding the pooled distribution, 0–1) by week for each SAP panel and density color. Across all SAPs, RTE increased across the monitoring period, with the highest density (2.228 × 10^6^) achieving elevated values earliest and maintaining them longest. In the Control panel, RTE at the highest density approached ~1.0 by mid-experiment and remained near the ceiling; the 228 × 10^3^ density rose steeply to ~0.95 by week 6, and 22.8 × 10^3^ reached ~0.8–0.85 by week 7. Zeba tracked the control closely: at the highest density, RTE rose rapidly to ~0.85–0.9 and remained high; at 228 × 10^3^, it climbed into the ~0.7–0.8 range by late weeks; at 22.8 × 10^3^, it increased steadily into the ~0.45–0.5 range. Stockosorb showed intermediate performance: high-density RTE reached ~0.9 by week 6; the mid-density rose to ~0.55; the low density to ~0.4. Aquaperla was generally the lowest: high-density RTE plateaued around ~0.65, the mid-density peaked near ~0.65 before softening late, and the low density ended near ~0.45–0.5. The significant SAP × Algae and SAP × Time interactions are thus reflected in diverging slopes and plateaus: Control and Zeba consistently occupy the upper envelope of RTE at a given density, Stockosorb is intermediate, and Aquaperla trails—particularly at the highest density where separation is most stable.

Descriptive RTE summaries reinforce these patterns (Table 2). For Control, average RTE (M ± SD; range) increased with density: 0.4 ± 0.4 (0.02–0.84) at 22.8 K, 0.6 ± 0.4 (0.10–0.94) at 228 K, and 0.9 ± 0.1 (0.74–0.98) at 2.228M. Zeba closely followed Control at each density—0.3 ± 0.1 (0.13–0.47), 0.5 ± 0.2 (0.21–0.77), and 0.8 ± 0.1 (0.52–0.87)—while Stockosorb was intermediate (0.2 ± 0.2 (0.03–0.43), 0.3 ± 0.2 (0.11–0.56), 0.7 ± 0.2 (0.36–0.91)). Aquaperla was the lowest of the four at each density (0.3 ± 0.2 (0.06–0.50), 0.4 ± 0.2 (0.13–0.66), 0.6 ± 0.1 (0.36–0.67)). Taken together, the RTE evidence indicates sustained and increasing absorbance over time across all gels, with Zeba performing comparably to Control (particularly at higher densities), Stockosorb generally intermediate, and Aquaperla lower—fully consistent with algae surviving and remaining active in the hydrogels, and with treatment- and density-specific dynamics over time.

#### 2.2.2. Chlorophyll Index

The repeated-measures model (Table 3) showed strong main effects of SAP treatment (ATS = 310.3, df = 2.87, *p* < 0.001), algae density (ATS = 1149.76, df = 1.88, *p* < 0.001), and time (ATS = 1159.66, df = 2.35, *p* < 0.001), with all interactions significant (SAP × Algae: ATS = 11.03, df = 4.93, *p* < 0.001; SAP × Time: ATS = 21.57, df = 4.43, *p* < 0.001; Algae × Time: ATS = 62.35, df = 4.18, *p* < 0.001; and SAP × Algae × Time: ATS = 17.19, df = 6.63, *p* < 0.001). Thus, chlorophyll-related signal increased over weeks and depended jointly on which gel was used and how many cells were present at the start.

The ChlIndex plots week-by-week in Figure 8 show that trajectories rose across all SAPs, with higher densities reaching higher plateaus earlier. In the Control panel, the highest density (2.228 × 10^6^) approached the ceiling (~0.95–1.0) by weeks 3–4 and remained high; the 228 × 10^3^ series climbed steeply, neared ~0.95 by weeks 5–6, and the 22.8 × 10^3^ series caught up late (~0.85–0.9 by week 7). Aquaperla consistently occupied the lower envelope: high density peaked near ~0.7 around week 6 with a slight late dip, mid-density topped out around ~0.6–0.65, and low density ended near ~0.5. Stockosorb showed intermediate performance, with high density reaching ~0.9 by week 6, mid-density ~0.6 by week 7, and low density ~0.45. Zeba also rose steadily: high density reached ~0.8–0.82 by weeks 4–5 before softening slightly, mid-density approached ~0.65, and low density finished near ~0.35. These shapes visually embody the significant SAP × Algae and SAP × Time interactions: separation among SAPs is clearest at the highest density and grows through time.

Descriptive RTE statistics corroborate the trajectories (Table 4). For Control, mean RTE (M ± SD; range) increased with density: 0.5 ± 0.4 (0.07–0.89) at 22.8 K, 0.6 ± 0.3 (0.20–0.96) at 228 K, and 0.9 ± 0.1 (0.78–0.98) at 2.228M. Zeba trailed Control but rose with density—0.2 ± 0.1 (0.06–0.36), 0.5 ± 0.2 (0.17–0.65), 0.7 ± 0.1 (0.46–0.82)—similar to Stockosorb (0.2 ± 0.2 (0.03–0.44), 0.4 ± 0.2 (0.11–0.60), 0.7 ± 0.2 (0.41–0.88)). Aquaperla remained lowest at each density (0.3 ± 0.2 (0.01–0.55), 0.4 ± 0.2 (0.12–0.66), 0.6 ± 0.1 (0.39–0.73)). Altogether, the RTE evidence shows robust time-dependent gains in chlorophyll index across all gels, with Control leading, Zeba and Stockosorb intermediate (approaching Control at higher densities), and Aquaperla consistently lower—a pattern consistent with algal persistence in hydrogels and density-dependent modulation of chlorophyll-related optics.

#### 2.2.3. Blue/Red Ratio

The repeated-measures model (Table 5) detected strong main effects of SAP treatment (ATS = 195.56, df = 2.48, *p* < 0.001), algae density (ATS = 41.93, df = 1.72, *p* < 0.001), and time (ATS = 89.28, df = 2.85, *p* < 0.001), with all interactions significant (SAP × Algae: ATS = 6.20, df = 3.49, *p* < 0.001; SAP × Time: ATS = 6.07, df = 4.47, *p* < 0.001; Algae × Time: ATS = 19.94, df = 4.26, *p* < 0.001; and SAP × Algae × Time: ATS = 2.99, df = 6.06, *p* = 0.006). Thus, the Blue/Red (BR) absorbance ratio evolved over weeks, and its trajectory depended jointly on the gel and initial biomass.

Figure 9 shows week-by-week Relative Treatment Effects by density color within each SAP panel in the case of Blue/Red_Ratio_. In Control, BR started high at low/mid-densities (weeks 1–2) and then declined steadily, converging near the lower end of the scale by weeks 6–7; the highest density dropped most quickly. Aquaperla and Stockosorb displayed a similar pattern—early peaks around weeks 2–3 followed by monotonic declines into late weeks—consistent with time-dependent shifts in spectral balance. In sharp contrast, Zeba maintained uniformly high BR throughout, with only modest declines for the low- and mid-density series and a temporary dip/recovery at the highest density. These shapes visually encode the significant SAP × Algae × Time interaction: BR ratio decays over time in Control/Aquaperla/Stockosorb, whereas Zeba sustains comparatively high BR levels across densities.

Descriptive RTE statistics underscore Zeba’s advantage (Table 6). For Zeba, mean RTEs were 0.9 (SD ≈ 0.0; range 0.87–0.99) at 22.8 K, 0.8 (SD ≈ 0.1; 0.71–0.96) at 228 K, and 0.7 (SD ≈ 0.1; 0.52–0.84) at 2.228M. By comparison, Control averaged 0.5 (0.05–0.92), 0.4 (0.08–0.79), and 0.2 (0.05–0.71) across the same densities; Aquaperla centered near 0.4 at all densities (ranges: 0.16–0.68, 0.22–0.58, 0.23–0.66); and Stockosorb averaged 0.5, 0.5, and 0.3 (ranges: 0.28–0.73, 0.18–0.70, 0.11–0.64). Together, these results indicate that, while BR generally shifts downward over time in most gels—consistent with evolving pigment optics—Zeba preserves a relatively high probability of exceeding the pooled distribution at all densities. This longitudinal pattern dovetails with the cross-sectional findings (Kruskal–Wallis + Dunn), in which Zeba frequently outperformed at least one comparator on BR.

#### 2.2.4. Normalized Difference Algal Index (NDAI)

The repeated-measures model (Table 7) showed significant main effects of SAP (ATS = 47.23, df = 1.58, *p* < 0.001), algae density (ATS = 66.07, df = 1.50, *p* < 0.001), and time (ATS = 99.94, df = 3.63, *p* < 0.001). Among interactions, SAP × Time (ATS = 3.39, df = 4.95, *p* = 0.005) and Algae × Time (ATS = 7.03, df = 5.25, *p* < 0.001) were significant, whereas SAP × Algae (ATS = 0.67, df = 2.28, *p* = 0.53) and the three-way SAP × Algae × Time (ATS = 1.64, df = 6.58, *p* = 0.124) were not. Thus, NDAI evolves over weeks, and the shape of the time course differs by SAP and by density, but the relative ordering of SAPs does not strongly depend on density, and there is no evidence of a joint SAP–density–time interaction.

NDAI plots (Figure 10) are showing that, across the study, Zeba maintains the highest RTE levels, Stockosorb and Aquaperla are intermediate, and Control is lowest, with the gap most apparent at the lowest starting density (22.8 × 10^3^) and still present at the mid-density (228 × 10^3^). At the highest density (2.228 × 10^6^), all trajectories shift downward relative to the pooled distribution, but Zeba remains on top, and Control remains lowest. The significant SAP × Time and Algae × Time interactions are visible as different slopes and plateaus across weeks for each gel and density, while the non-significant SAP × Algae term matches the visual impression that the ranking among gels is broadly consistent across densities.

For Zeba (Table 8), mean RTEs (M ± SD; range) were 0.8 ± 0.1 (0.69–0.94) at 22.8 K, 0.7 ± 0.1 (0.52–0.85) at 228 K, and 0.5 ± 0.1 (0.36–0.60) at 2.228M. Stockosorb and Aquaperla followed a similar density-dependent decline (Stockosorb: 0.7 ± 0.2 (0.44–0.96), 0.5 ± 0.2 (0.32–0.89), 0.3 ± 0.1 (0.19–0.58); Aquaperla: 0.7 ± 0.3 (0.31–0.98), 0.5 ± 0.2 (0.31–0.90), 0.4 ± 0.1 (0.16–0.53)). Control was consistently lowest (0.4 ± 0.4 (0.08–0.89), 0.3 ± 0.3 (0.04–0.72), 0.2 ± 0.1 (0.09–0.34)). Because higher RTE indicates a greater probability that observations exceed the pooled distribution (i.e., less-negative NDAI values in our scaling), these longitudinal results show a persistent advantage for Zeba over time, with Stockosorb/Aquaperla intermediate and Control lowest, and with all gels showing lower NDAI RTE at higher starting densities. This pattern is coherent with the cross-sectional analysis, where Zeba exceeded Control at the lowest and highest densities, and indicates that algae remain active in gels while NDAI reflects density- and time-dependent spectral changes.

#### 2.2.5. Photochemical Reflectance Index (PRI)

The repeated-measures model (Table 9) detected clear main effects of SAP treatment (ATS = 120.57, df = 2.75, *p* < 0.001), algae density (ATS = 43.24, df = 1.69, *p* < 0.001), and time (ATS = 9.53, df = 4.29, *p* < 0.001). All interactions were significant as well (SAP × Algae: ATS = 14.90, df = 4.02, *p* < 0.001; SAP × Time: ATS = 3.69, df = 6.00, *p* = 0.001; Algae × Time: ATS = 18.51, df = 5.59, *p* < 0.001; SAP × Algae × Time: ATS = 3.84, df = 6.67, *p* < 0.001), indicating that PRI trajectories over weeks depend jointly on gel type and starting biomass, with patterns that differ among SAPs and across densities.

Figure 11 shows that, across SAPs, PRI RTE generally increases from early to mid-experiment and then stabilizes at modest positive levels, consistent with the significant effect of time. Separation by density is SAP-specific: in Stockosorb, the highest density (2.228 × 10^6^) rises earlier and remains above the lower densities, while, in Control and Zeba, the density curves largely run in parallel with only small gaps; Aquaperla exhibits shallow slopes with limited separation among densities. These visual features mirror the significant SAP × Algae and SAP × Algae × Time terms, i.e., density effects on PRI are contingent on the gel and evolve over time.

PRI trajectories generally rise from early to mid-experiment and then stabilize at modest positive levels. Density separation is SAP-specific: it is most pronounced in Stockosorb, where the highest density (2.228 × 10^6^) remains above the lower densities, while Control, Aquaperla, and Zeba display largely parallel curves with small gaps. These patterns visualize the significant SAP × Algae and SAP × Time effects from the omnibus tests and the localized density advantage under Stockosorb.

To localize the SAP × Algae interaction for PRI, we compared algae densities within each SAP using Wilcoxon rank-sum tests on the average PRI across weeks 2–6, controlling the false discovery rate with Benjamini–Hochberg Procedure (Table 10). Only Stockosorb showed significant separation: 2.228 × 10^6^ > 22.8 × 10^3^ (mean difference = 0.017, *p*_BH < 0.05) and 2.228 × 10^6^ > 228 × 10^3^ (mean difference = 0.014, *p*_BH < 0.05); all density contrasts were non-significant within Control, Aquaperla, and Zeba. Thus, the global SAP × Algae and higher-order interaction for PRI is primarily driven by Stockosorb, where higher biomass yields consistently higher PRI, whereas density effects in the other gels are weak or transient. Overall, the PRI time course indicates gradual, SAP-specific gains over time with a clear density advantage only under Stockosorb, consistent with the cross-sectional result that PRI differences are most pronounced at intermediate density.

#### 2.2.6. Normalized Phaeophytinization Index (NPQI)

The repeated-measures model (Table 11) showed strong main effects of SAP (ATS = 133.37, df = 2.35, *p* < 0.001), algae density (ATS = 113.14, df = 1.76, *p* < 0.001), and time (ATS = 13.21, df = 3.09, *p* < 0.001), with significant SAP × Algae (ATS = 4.76, df = 3.78, *p* = 0.001), SAP × Time (ATS = 2.37, df = 6.48, *p* = 0.023), and Algae × Time (ATS = 4.13, df = 4.97, *p* = 0.001) interactions; the three-way term was borderline (ATS = 1.89, df = 8.70, *p* = 0.05), indicating gel- and density-dependent time courses with hints of higher-order modulation.

Figure 12 shows distinct patterns by SAPs. In Control, NPQI RTE declines steadily at all densities, falling from higher values in week 1 to a low plateau by weeks 5–7. Aquaperla and Stockosorb exhibit early peaks (especially at low/mid densities) followed by declines; in Aquaperla, the highest density rises slowly from low values to meet the mid-density late, whereas in Stockosorb the ranking is stable (22.8 × 10^3^ > 228 × 10^3^ > 2.228 × 10^6^) across weeks. Zeba maintains uniformly high RTE with only modest erosion over time at low/mid densities and a dip–recovery pattern at the highest density—visually reflecting the significant SAP × Algae and SAP × Time interactions.

Control shows a steady decline in NPQI RTE across densities; Aquaperla and Stockosorb peak early (weeks 2–3) and then decrease; Zeba maintains comparatively high RTE with only modest erosion over time. Density gradients are most pronounced for Stockosorb and Zeba (generally 22.8 × 10^3^ > 228 × 10^3^ > 2.228 × 10^6^), illustrating the significant SAP × Algae and SAP × Time dependencies observed in the omnibus tests.

Based on the Wilcoxon rank-sum test (Table 12), control shows no significant density separations. In Aquaperla, the highest density (2.228 × 10^6^) is lower than 22.8 × 10^3^ and lower than 228 × 10^3^ (mean differences = 0.070 and 0.048; both *p* < 0.05). In Stockosorb, all density pairs differ (0.034, 0.059, 0.025; all *p* < 0.05) with the same ordering 22.8 × 10^3^ > 228 × 10^3^ > 2.228 × 10^6^. In Zeba, all pairs also differ (0.037, 0.072, 0.035; all *p* < 0.05), again showing a graded decline with increasing density. These contrasts confirm that the density dependence of NPQI is gel-specific and strongest for Stockosorb and Zeba, modest for Aquaperla (driven by the high-density series), and negligible for Control—coherent with the cross-sectional result that Zeba exceeded Control across densities, while Aquaperla only surpassed Control at the highest density.

### 2.3. Results of PCA

We found that PC1 alone captures ~97% of the total variance in our absorbance spectra (Table 13), confirming that overall pigment concentration (algal biomass) is the dominant signal. PC2 accounts for only ~1.4–1.6% of the variance and captures first-order, treatment-specific spectral shape differences. Subsequent components each explain less than 0.2% of the variance, so we focus our interpretation on PC1 (overall pigment concentration) and PC2 (treatment-specific spectral shape differences) rather than presenting the full scree plot (Table 1). Variance explained by the first five principal components. PC1 accounts for the vast majority of spectral variance (>97%), PC2 captures the minor, treatment-dependent variation (~1.4%), and PCs 3–5 each contribute ≤0.65% of the variance.

The PCA score plot (Figure 13) of raw absorbance spectra reveals two main axes of variation in our dataset. The first principal component (PC1), which accounts for 97.7% of the total variance, runs horizontally and establishes a clear biomass gradient: samples collected at low cell densities (22.8 K cells/mL) cluster tightly on the left side of the plot, whereas those at intermediate (228 K cells/mL) and high (2228 K cells/mL) concentrations migrate progressively toward the right. This confirms that overall pigment (chlorophyll) load is overwhelmingly the dominant factor shaping the absorbance spectra. The second principal component (PC2), capturing 1.4% of the variance and plotted on the vertical axis, separates samples by SAP treatment at each density level. Within each density band—denoted by distinct point shapes—Control, Aquaperla, Stockosorb, and Zeba treatments occupy subtly different PC2 positions, indicating that each polymer modulates the relative spectral shape beyond simple changes in biomass. For instance, at medium density (228 K cells/mL), Aquaperla-treated samples tend to lie above the midline while Stockosorb-treated samples fall below, suggesting treatment-specific shifts in accessory-pigment absorbance or scattering properties in the green–yellow region. The combined use of color and shape coding allows us to visualize these overlapping clusters clearly, demonstrating that, while PC1 captures the dominant concentration effect, PC2 successfully isolates the more nuanced, treatment-dependent spectral signatures. This multivariate framework thus both validates our choice of indices targeting chlorophyll peaks for biomass estimation and highlights the importance of blue/red versus green–yellow contrasts for discriminating SAP-induced spectral changes.

The PCA loading curves (Figure 14) furnish a rigorous, data-driven rationale for why our six indices collectively capture both the overwhelmingly large pigment signal and the fine-scale, treatment-specific spectral shifts induced by the four superabsorbent polymers. PC1, which explains nearly 97% of the total variance, exhibits strong positive loadings at the two classic chlorophyll-*a* absorption bands (approximately 430–450 nm and 670–680 nm) and remains positive across much of the visible spectrum; this demonstrates that total pigment concentration dominates our absorbance data, a pattern we directly quantify with Integrated Absorbance (which sums the entire spectral envelope) and the Chlorophyll Index (which contrasts the blue and red peaks against adjacent baselines). In contrast, PC2—though responsible for only about 1–2% of variance—is essential for teasing apart the subtler ways each SAP perturbs the spectral shape once biomass is controlled. PC2’s loading vector plunges negative in those same blue and red chlorophyll regions but then rises sharply through the green–yellow shoulder (roughly 525–600 nm), signaling that treatment effects manifest as relative enhancement of accessory-pigment absorbance or altered scattering in that mid-spectral window.

To track those nuanced features, we selected two “shape” indices—the Blue/Red Ratio and the Normalized Difference Algal Index (NDAI)—which directly measure the relative strength of the blue versus red shoulders where PC2 loadings are most negative. However, because PC2 loadings peak positively in the green–yellow region, we also needed metrics tuned precisely to that wavelength band. The Photochemical Reflectance Index (PRI) fulfills this role by comparing absorbance (or reflectance) at ~531 nm—where xanthophyll-cycle pigments and carotenoids absorb strongly—to a reference wavelength around ~570 nm that is minimally influenced by chlorophyll. Elevations in PRI, therefore, signal increases in carotenoid content or light-scattering changes that correspond exactly to PC2’s positive loading bump. Complementing PRI, the Pigment-to-Carotenoid Index (NPQI, sometimes referred to as the PSRI) contrasts the green–yellow shoulder (usually near 550 nm) against the red chlorophyll peak (~680 nm), effectively isolating shifts in the carotenoid-to-chlorophyll ratio. When NPQI rises, it indicates a treatment-driven reallocation of pigment composition away from chlorophyll and toward photoprotective carotenoids—again mirroring the sign and magnitude of PC2 in our loading plot.

By selecting Integrated Absorbance and the Chlorophyll Index for PC1 and pairing BR_Ratio_, NDAI, PRI, and NPQI for PC2—two of which target the blue/red contrast and two focusing on the green–yellow shoulder—we ensure that our entire index suite spans the full multivariate fingerprint unveiled by PCA. This disciplined alignment between loadings and indices not only maximizes our ability to quantify total biomass and polymer-specific effects but also provides a transparent, hypothesis-free justification for each index’s inclusion in downstream statistical and graphical analyses.

## 3. Discussion

This study provides the first comprehensive evaluation of how different superabsorbent polymer (SAP) chemistries influence the growth and pigment composition of freely suspended *Chlorella vulgaris* across a range of initial cell densities. Our findings demonstrate that SAPs do not eliminate microbial growth but modulate biomass accumulation and pigment-related spectral indices in a density- and chemistry-dependent manner. This supports our hypothesis that hydrogels can alter physiological responses through changes in nutrient availability, hydration stability, and light transmission, without universally suppressing metabolic activity.

### 3.1. SAP-Specific Modulation of Biomass-Related Indices

Integrated Absorbance and the Chlorophyll Index—both primarily reflecting total pigment content—were highest in the Control treatment, with Zeba often matching Control performance across densities, Stockosorb generally intermediate, and Aquaperla consistently lower. These results were consistent across cross-sectional and longitudinal analyses, and the PCA confirmed that these indices are dominated by the first principal component (PC1), which explained ~97% of spectral variance and was strongly associated with chlorophyll-*a* absorption peaks.

This means that, relative to the untreated Control, Aquaperla frequently suppressed biomass accumulation (especially at high density), Stockosorb produced intermediate values that were below Control but not as strongly reduced as Aquaperla, while Zeba maintained biomass levels that were statistically indistinguishable from Control. Thus, Zeba emerges as the only polymer that consistently sustained growth comparable to the baseline, whereas Aquaperla and Stockosorb both imposed varying degrees of reduction.

The comparable performance of Zeba to Control suggests that starch-based polyacrylamide hydrogels can maintain biomass levels similar to untreated cultures, potentially due to their biodegradability, nutrient exchange capacity, and relatively low interference with light penetration [9,40]. In contrast, the lower values observed with Aquaperla may reflect ionic imbalances or osmotic stress caused by strong cation exchange properties [11,13]. Future mechanistic studies measuring dissolved nutrient concentrations and ionic composition in the growth medium would help clarify these effects. Because we did not measure dissolved nutrients, ion concentrations, or hydrogel stability directly, our interpretations regarding these mechanisms remain hypothetical. Likewise, we cannot claim effects on metabolic activity, as transcriptomic or metabolomic data were not collected. Future studies should address these processes explicitly to confirm the physiological basis of the observed pigment responses.

### 3.2. Accessory Pigment Responses and Spectral Shape Shifts

Indices sensitive to accessory pigments—Blue/Red Ratio, NDAI, PRI, and NPQI—were more variable among SAPs, aligning with PCA’s PC2 (1.4% variance), which captured green–yellow spectral shifts. Zeba consistently outperformed Control in Blue/Red Ratio, NDAI, and NPQI. These results indicate a greater relative contribution of carotenoids and xanthophylls, pigments associated with photoprotection and stress mitigation [32,38]. Biologically, elevated Blue/Red ratios typically signal increased carotenoid relative to chlorophyll content, a hallmark of photoprotective responses [41,42,43]. Thus, Zeba’s consistent elevation of BR values suggests mild stress priming, where algae allocate more resources to carotenoids for protective roles while still sustaining growth. Elevated Blue/Red ratios generally signify a relative increase in carotenoid absorption compared to chlorophyll, reflecting enhanced photoprotective capacity. In algal cultures, this shift suggests a trade-off, where resources are reallocated from growth-oriented chlorophyll synthesis toward stress-mitigation pathways, enabling cells to maintain resilience under variable light and osmotic conditions [41,43].

This effect was density-dependent, being most pronounced at low and high starting cell concentrations for NDAI and across all densities for NPQI. Sustained high BR and NPQI values in Zeba treatments suggest that this polymer may create microenvironments that trigger moderate stress responses, resulting in pigment reallocation without biomass loss—potentially advantageous for enhancing high-value pigment production in biotechnology applications.

Stockosorb (γ-polyglutamate) generally showed intermediate performance on pigment composition indices, consistent with its biocompatibility and mild nutrient buffering capacity [7]. Aquaperla occasionally showed higher NPQI at high densities but otherwise lagged in pigment-related metrics, reinforcing its comparatively less favorable performance for sustained biomass productivity.

It is important to note that both PRI and NPQI were originally developed for vascular plants. Nonetheless, several studies have successfully adapted these indices for algal systems, demonstrating their value in tracking xanthophyll- and carotenoid-related responses in microalgae [35,38]. While no mathematical correction was applied here, the consistent treatment-specific trends we observed indicate that these indices can still provide meaningful insights into pigment reallocations in *Chlorella vulgaris*.

The observed differences among SAP chemistries likely arise from their physicochemical interactions with the growth medium. Aquaperla may alter ionic strength and osmotic balance, creating stress that reduces biomass accumulation [44], a phenomenon consistent with broader evidence of ionic effects from synthetic SAPs [45]. In contrast, starch-based Zeba may act as both a water reservoir and a slow-release carbon source, which could explain its ability to maintain control-like biomass while increasing carotenoid-related indices [46]. The Stockosorb is known for its biocompatibility and mild buffering capacity, which may explain its intermediate performance [47,48]. These mechanisms suggest that polymer chemistry not only influences plant–water relations in soils but can also directly modulate microbial physiology in aqueous systems [49,50].

### 3.3. Density-Dependent Effects and Interaction Patterns

Initial biomass strongly shaped outcomes, with higher starting densities typically achieving higher integrated absorbance and chlorophyll index values more rapidly, as reflected in the steep early RTE increases. However, pigment composition indices sometimes showed the opposite pattern, with low-density cultures exhibiting relatively higher carotenoid allocation. This aligns with literature showing that low-density photoautotrophic cultures experience greater light exposure per cell, potentially triggering photoprotective pigment synthesis [8,31,51]. The significant SAP × Density × Time interactions for several indices highlight that polymer effects cannot be generalized across biomass levels—a consideration critical for designing microbial–SAP co-culture systems.

### 3.4. Implications for Hydrogel–Microbe Applications

Our results extend previous work on immobilized algal–hydrogel systems [8,30,31] by showing that free-living microorganisms can maintain or adjust growth and pigment profiles in the presence of hydrogels, with responses dependent on polymer chemistry. While this study used *C. vulgaris* as a model organism, the patterns observed—such as Zeba’s ability to maintain biomass while enhancing accessory pigment indices—may be relevant to other beneficial microorganisms used in agriculture, including photosynthetic *cyanobacteria* and plant-growth-promoting fungi such as *Trichoderma* spp. [52]. These organisms contribute to nutrient cycling, plant resilience, and soil health, and understanding their compatibility with SAPs could guide applications in biofertilization, integrated crop management, and sustainable biotechnological production systems.

These patterns have practical importance for biotechnology. The ability of Zeba to preserve biomass while enhancing carotenoid allocation indicates that biodegradable hydrogels may be co-opted to stimulate pigment production in algal cultivation systems without compromising yield. These responses could be harnessed for the commercial production of lutein, zeaxanthin, and other valuable carotenoids. In contrast, Aquaperla’s suppressive effects highlight the need for careful SAP selection in soil–microbe systems to prevent unintended growth inhibition. Stockosorb’s intermediate performance suggests it may serve as a neutral or modulatory matrix in biotechnological setups. Together, these findings underscore that hydrogel chemistry is a key design variable for both algal biotechnology and agricultural biofertilization strategies.

### 3.5. Limitations and Future Directions

This study was conducted under constant temperature, light, and nutrient conditions; therefore, results may differ under fluctuating environmental regimes or in nutrient-limited systems. Direct measurements of nutrient dynamics, pH shifts, and dissolved organic carbon in SAP treatments would help elucidate the mechanisms behind the observed spectral shifts. Additionally, targeted pigment or metabolite analysis (e.g., HPLC for pigments, GC–MS for secondary metabolites) could validate interpretations of spectral indices in both algae and other microbial taxa. Expanding this research to include cyanobacteria, *Trichoderma*, and other agriculturally relevant microbes will provide a more complete understanding of SAP–microbe interactions, enabling tailored polymer–microbe combinations for both environmental safety and production efficiency.

## 4. Materials and Methods

### 4.1. Experimental Design and Algal Culture

This study examined the effects of three commercially available superabsorbent polymers (SAPs or hydrogels) on the growth of *Chlorella vulgaris* Beijerinck under controlled laboratory conditions. Three commercially available superabsorbent polymers (SAPs) were tested: Aquaperla^®^ (potassium polyacrylate, De Ceuster Meststoffen nv, Grobbendonk, Belgium), Zeba Plus SP^®^ (starch-based polyacrylamide-co-acrylic acid potassium salt, UPL Holdings Coöperatief, Amsterdam, The Netherlands), and Stockosorb^®^ 660 Medium (γ-polyglutamate, Stockosorb Degussa GmbH, Frankfurt, Germany). Hereafter referred to as Aquaperla, Zeba, and Stockosorb, respectively, in addition to an untreated control.

SAPs were added in sterile, as-received form from the suppliers, without further sterilization. While sterile handling minimized contamination, we cannot fully exclude the presence of background microorganisms introduced with the polymers.

Three initial algal cell densities were tested: 22.8 × 10^3^, 228 × 10^3^, and 2.228 × 10^6^ cells/mL (hereafter 22.8 K, 228 K, and 2228 K). Each SAP–algae combination was prepared in four replicates, resulting in 48 experimental units. The experimental units consisted of sterile Petri dishes containing 32 mL of algal suspension prepared at the designated concentration and pH with 0.88 g of SAP per Petri dish from Aquaperla and Zeba, and 0.6 g from Stockosorb, respectively.

The *C. vulgaris* culture was maintained in Zhender-8 medium (pH 6–7) [53,54] before the experiment. The initial physicochemical properties of the algal suspensions were measured before SAP addition:22.8 K: pH 7.0, EC 0.7835 mS cm^−1^.228 K: pH 7.0, EC 0.7725 mS cm^−1^.2228 K: pH 8.0, EC 0.7785 mS cm^−1^.

### 4.2. Growth Conditions and Measurements

The in vitro trial was conducted in an ICO105 growth and climate chamber (Memmert GmbH, Schwabach, Germany) under controlled conditions of 24 °C, 70% relative humidity, and a 16 h light/8 h dark photoperiod for seven weeks during the first half of 2025 at the Laboratory of the Institute of Agronomy, Hungarian University of Agriculture and Life Sciences. The average photosynthetic photon flux density (PPFD) provided by the built-in LED system was 75.8 µmol m^−2^ s^−1^. The PPFD was measured by a hand-held spectroradiometer (Mavospec Base, GOSSEN Foto—und Lichtmesstechnik GmbH, Nürnberg, Germany).

This intensity is within the optimal range reported for *Chlorella vulgaris* cultivation. The experimental setup is shown in Table 14.

Spectral transmittance of each sample was recorded once per week, and the data were used to calculate spectral indices related to algal growth and stress status. In addition, algal biomass was quantified by measuring the optical density (OD) of the suspensions at 750 nm using a UV/VIS spectrophotometer (Figure 15) (Camspec M330, single beam; Camspec, Crawley, UK). Optical density was measured in triplicate for each sample to ensure analytical precision.

### 4.3. Spectral Data Processing and Calculation of Integrated Absorbance

Light transmittance spectra (I_λ_) in the wavelength range 380–780 nm were recorded for each Petri dish using a spectroradiometer (Mavospec Base, GOSSEN Foto—und Lichtmesstechnik GmbH, Nürnberg, Germany) under standardized illumination. For each measurement date, a reference spectrum (I_0,λ_) was acquired by measuring the incident light without Petri dishes. The absorbance spectrum (A_λ_) was calculated according to the Beer–Lambert law [55]:(1)$$A_{\lambda }=-{log}_{10}\left(\frac{I_{\lambda }}{I_{0,\lambda }}\right)$$
where Iλ is the transmitted light intensity through the sample at wavelength λ, and I0,λ is the reference intensity.

### 4.4. Calculated Indicies

#### 4.4.1. Integrated Absorbance Calculation

The Integrated Absorbance (IA), Ref. [56] representing the total absorption across the photosynthetically active spectrum, was computed as the area under the absorbance curve between 400 and 700 nm:(2)$$IA=\int_{400}^{700}A_{\lambda }d\lambda$$

Numerical integration was performed using the trapezoidal rule [57]:(3)$$IA\approx \sum_{i=1}^{n-1}\frac{A_{\lambda_{i}}+A_{\lambda_{i+1}}}{2}\left(\lambda_{i+1}-\lambda_{i}\right)$$
where A_λi_ is the absorbance at wavelength λi, and n is the number of measured spectral points.

The Integrated Absorbance reflects the overall light absorption by pigments within the algal suspension. Higher values indicate greater pigment density and algal biomass, whereas lower values suggest reduced photosynthetic capacity or growth inhibition. Because absorbance is dimensionless, Integrated Absorbance is reported in arbitrary units (AUs) suitable for relative comparisons among treatments.

#### 4.4.2. Calculation of Chlorophyll Index (ChlIndex)

In addition to the integrated absorbance, a Chlorophyll Index (ChlIndex) [58] was calculated to specifically quantify the relative absorption of chlorophyll pigments. Chlorophylls absorb strongly in the red region (~680 nm) while exhibiting minimal absorption in the near-infrared region (~750 nm). Therefore, the index was defined as the difference in absorbance between these two characteristic wavelengths:(4)$$ChlIndex=A_{680}-A_{750}$$
where A680 and A750 are the absorbance values at 680 nm and 750 nm, respectively.

A_680_ corresponds to the chlorophyll a absorption peak, directly related to pigment concentration and photosynthetic capacity. A_750_ is a reference wavelength with minimal pigment absorption, accounting for baseline scattering effects. Therefore, ChlIndex reflects the relative abundance of chlorophyll pigments and serves as an indicator of algal photosynthetic activity.

#### 4.4.3. Calculation of Blue/Red Absorbance Ratio (BR_Ratio_)

To assess changes in pigment composition and spectral absorption characteristics of the algal cultures, the Blue/Red Absorbance Ratio (BR_Ratio_) [59] was calculated from the absorbance spectrum. This index compares absorption in the blue region (~440 nm), dominated by chlorophyll *a* and accessory pigments, to the red region (~680 nm), where chlorophyll *a* has a major absorption peak:(5)$$BR_{Ratio}=\frac{A_{440}}{A_{680}}$$
where A_440_ and A_680_ are the absorbance values measured at 440 nm and 680 nm, respectively.

The blue region (440~440 nm) is influenced by chlorophyll *a* and carotenoids, which also absorb strongly in this range. The red region (680~680 nm) corresponds to the primary absorption peak of chlorophyll-*a* and is strongly correlated with photosynthetic pigment content. An increase in BR_Ratio_ suggests a relative enhancement in blue light absorption, potentially due to changes in pigment composition (e.g., carotenoid accumulation), whereas a decrease indicates stronger dominance of red absorption. Thus, BR_Ratio_ provides insights into pigment dynamics and algal physiological status under different treatments.

#### 4.4.4. Calculation of Normalized Difference Algal Index (NDAI)

The Normalized Difference Algal Index (NDAI) [60] was computed to provide a normalized measure of algal pigment absorption by comparing spectral reflectance (or absorbance) in the red and near-infrared regions. This index is analogous to vegetation indices (e.g., NDVI) but adapted for algal suspensions. The NDAI was calculated as:(6)$$NDAI=\frac{A_{680}-A_{750}}{A_{680}+A_{750}}$$
where A_680_ and A_750_ are the absorbance values at 680 nm (chlorophyll-*a* peak) and 750 nm (reference wavelength with minimal pigment absorption), respectively.

The red absorption at 680 nm is strongly associated with chlorophyll-*a* content. The near-infrared baseline at 750 nm reflects light scattering and serves as a non-absorbing reference point. NDAI values range between −1 and 1, providing a dimensionless, normalized indicator that is less sensitive to absolute light intensity. Higher NDAI values indicate strong pigment absorption and higher algal biomass, whereas lower values suggest pigment degradation or reduced algal density. Thus, NDAI serves as a robust index to compare algal physiological responses across treatments while minimizing the influence of external illumination differences.

#### 4.4.5. Calculation of Photochemical Reflectance Index (PRI)

The Photochemical Reflectance Index (PRI) [61,62] was computed to quantify relative changes in xanthophyll-cycle pigments and carotenoid content in the algal cultures. PRI is defined as the normalized difference between reflectance (or, in our case, absorbance) at 531 nm—where xanthophyll and carotenoids absorb strongly—and at a reference wavelength of 570 nm, which lies outside major chlorophyll absorption features:(7)$$PRI=\frac{A_{530}-A_{570}}{A_{530}+A_{570}}$$
where A_530_ and A_570_ are the absorbance values at 530 nm and 570 nm, respectively.

Because carotenoids and xanthophyll-cycle pigments absorb in the green–yellow region (~531 nm) while chlorophyll absorbance is minimal at 570 nm, an increase in PRI indicates a relative enrichment of photoprotective pigments or changes in light-scattering properties. This index thus provides a sensitive measure of treatment-induced shifts in accessory-pigment pools and photosynthetic regulation.

#### 4.4.6. Calculation of Normalized Pigment-to-Carotenoid Index (NPQI)

To track relative changes between chlorophyll and carotenoid pools—particularly the treatment-specific effects that emerged in PC2 of our PCA—we calculated the Normalized Pigment-to-Carotenoid Index (NPQI) [63] as follows:(8)$$NPQI=\frac{A_{550}-A_{680}}{A_{550}+A_{680}}$$
where A_550_ is the absorbance at 550 nm, capturing the broad carotenoid absorption shoulder, and A_680_ is the absorbance at 680 nm, corresponding to the primary chlorophyll-*a* absorption peak. NPQI therefore expresses the carotenoid-to-chlorophyll ratio in a single metric. Values approaching +1 indicate a dominance of carotenoid absorption (e.g., treatment-induced photoprotective pigment accumulation), whereas values near −1 reflect spectra dominated by chlorophyll-*a*.

### 4.5. Statistical Analysis

All statistical analyses were conducted in R version 4.2.0 (R Core Team, 2022). Data were first tested for normality and homogeneity of variances using the Shapiro–Wilk test and Levene’s test, respectively. Both tests indicated significant deviations from normality (*p* < 0.001) across all variables and groups, as well as significant heterogeneity of variances (*p* < 0.001). Consequently, the data violated the assumptions for parametric repeated-measures ANOVA, and a rank-based nonparametric approach was adopted.

A Principal Component Analysis (PCA) was conducted to explore multivariate patterns among the spectral indices across treatments and algae densities. The analysis was performed on standardized data (z-scores) and visualized where PCA biplots illustrated relationships among variables and clustering of treatment groups, aiding in the identification of the main gradients in the dataset. Based on the results of the PCA, we chose the six examined indices.

After a Kruskal–Wallis non-parametric analysis of the chosen indices, a pairwise post hoc comparisons between SAP treatments within each algae density were performed using Dunn’s test with Holm–Bonferroni correction. Group differences were illustrated via letter and displayed above the boxplots. Descriptive statistics (mean ± standard deviation, median, range) were calculated for each treatment and algae density.

For the spectral indices (IntegratedAbs, ChlIndex, BR_Ratio, NDAI, PRI, NPQI), a nonparametric repeated-measures factorial design was applied using the *nparLD* package by [64]. This method is robust to non-normality, variance heterogeneity, and unbalanced designs. The model included two between-subject factors—SAP treatment (Control, Aquaperla, Stockosorb, Zeba) and Algae density (22.8 × 10^3^, 228 × 10^3^, 2.228 × 10^6^ cells/mL)—and one within-subject factor, Time (weeks 2–7). The ANOVA-Type Statistic (ATS) was used to evaluate main effects and interactions, with degrees of freedom estimated via the Box approximation. Significance was determined at α = 0.05. Relative Treatment Effects (RTEs) were calculated for each factor–level combination to quantify the probability that a randomly selected observation from a given level exceeds one from the reference level. RTE values above 0.5 indicate higher index values relative to the reference. After, the temporal dynamics of RTEs were visualized.

## 5. Conclusions

This study demonstrates that superabsorbent polymers (SAPs) influence *Chlorella vulgaris* growth and pigment composition with treatment- and density-dependent effects and without evidence of complete growth inhibition. The starch-based polyacrylamide (Zeba) consistently matched or approached Control biomass indices while enhancing carotenoid- and xanthophyll-sensitive metrics (Blue/Red Ratio, NDAI, NPQI), suggesting that it can maintain yields while promoting pigment reallocations beneficial for photoprotection or high-value compound production. In contrast, Aquaperla frequently reduced biomass-related measures—especially at high densities—while Stockosorb produced intermediate responses.

Initial cell density strongly shaped both growth trajectories and pigment allocation: higher densities generally accelerated biomass accumulation, whereas lower densities often promoted greater relative carotenoid allocation. These patterns have dual relevance: (i) in biotechnology, for selecting hydrogel–microbe combinations that optimize biomass and metabolite outputs and (ii) in environmental contexts, for assessing how residual SAPs may selectively influence aquatic and soil microbial communities.

Although this work focused on *C. vulgaris*, extending these experiments to agriculturally relevant microorganisms—such as photosynthetic cyanobacteria and plant-growth-promoting fungi (*Trichoderma* spp.)—will clarify whether the observed polymer–microbe interactions generalize across functional groups, providing a broader foundation for applying SAPs in integrated crop management, biofertilization, and sustainable biotechnological systems.

## Figures and Tables

**Figure 1 plants-14-02962-f001:**
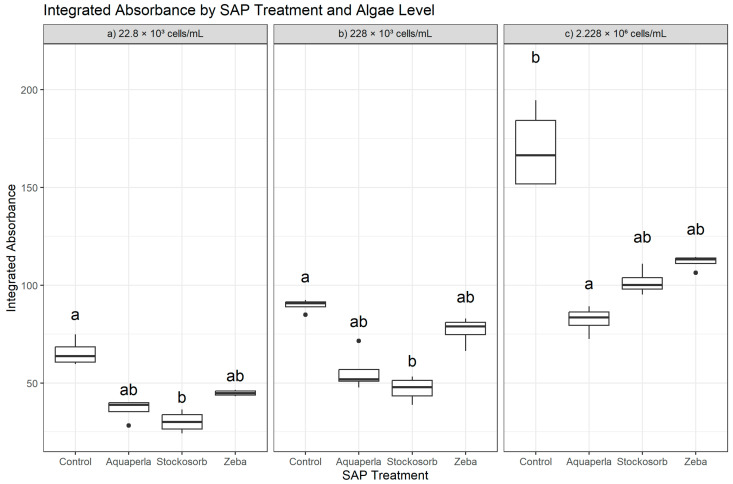
Cross-sectional Integrated Absorbance by SAP within algal densities (averaged over weeks 2–7). Boxes show the median and interquartile range; whiskers extend to 1.5 × IQR; points are individual observations. Panels correspond to algal densities ((**a**): 22.8 × 10^3^; (**b**): 228 × 10^3^; (**c**): 2.228 × 10^6^ cells/mL). Letters above boxes denote groups not significantly different at α = 0.05 based on Dunn–Bonferroni pairwise tests following a Kruskal–Wallis test conducted separately within each density.

**Figure 2 plants-14-02962-f002:**
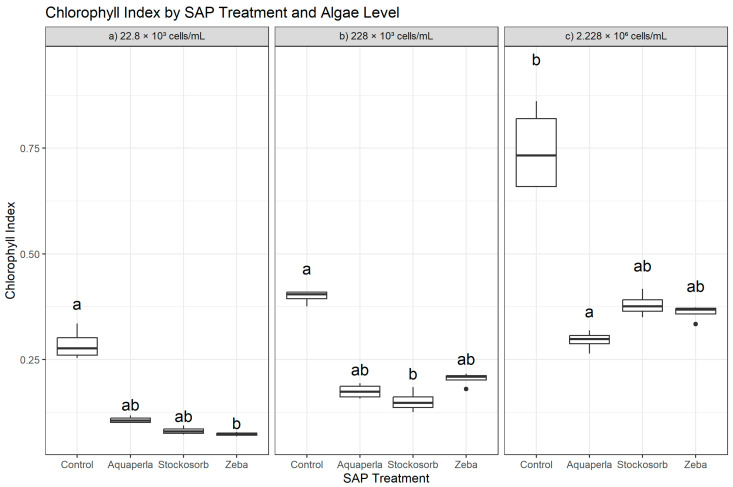
Cross-sectional Chlorophyll Index by SAP within algal densities (averaged over weeks 2–7). Boxes show median and interquartile range; whiskers extend to 1.5 × IQR; points are individual observations. Panels correspond to algal densities ((**a**): 22.8 × 10^3^; (**b**): 228 × 10^3^; (**c**): 2.228 × 10^6^ cells/mL). Letters above boxes denote groups not significantly different at α = 0.05 by Dunn–Bonferroni pairwise tests following a Kruskal–Wallis test run separately within each density.

**Figure 3 plants-14-02962-f003:**
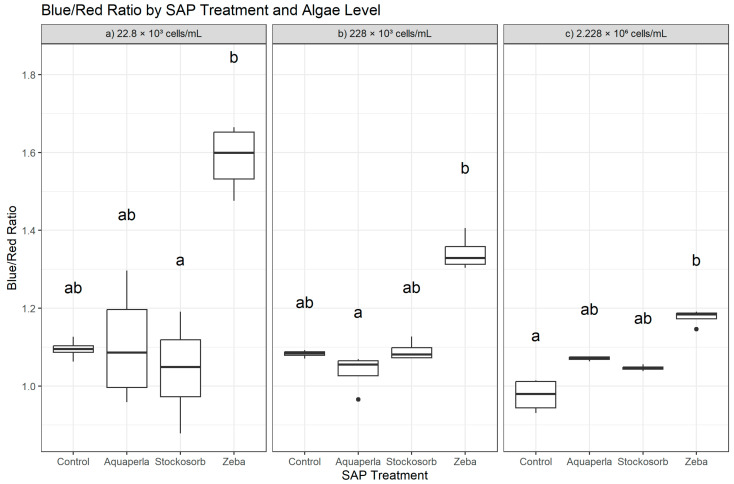
Cross-sectional Blue/Red Ratio by SAP within algal densities (averaged over weeks 2–7). Boxes show the median and interquartile range; whiskers extend to 1.5 × IQR; points are individual observations. Panels correspond to algal densities ((**a**): 22.8 × 10^3^; (**b**): 228 × 10^3^; (**c**): 2.228 × 10^6^ cells/mL). Letters above boxes denote groups not significantly different at α = 0.05 by Dunn–Bonferroni tests following density-specific Kruskal–Wallis analyses.

**Figure 4 plants-14-02962-f004:**
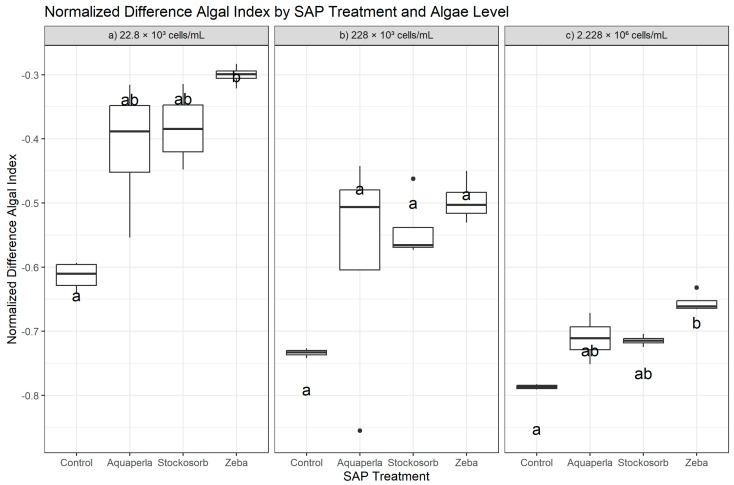
Cross-sectional NDAI by SAP within algal densities (averaged over weeks 2–7). Boxes show the median and interquartile range; whiskers extend to 1.5 × IQR; points are individual observations. Panels correspond to algal densities ((**a**): 22.8 × 10^3^; (**b**): 228 × 10^3^; (**c**): 2.228 × 10^6^ cells/mL). Letters above boxes denote groups not significantly different at α = 0.05 by Dunn–Bonferroni tests following density-specific Kruskal–Wallis analyses.

**Figure 5 plants-14-02962-f005:**
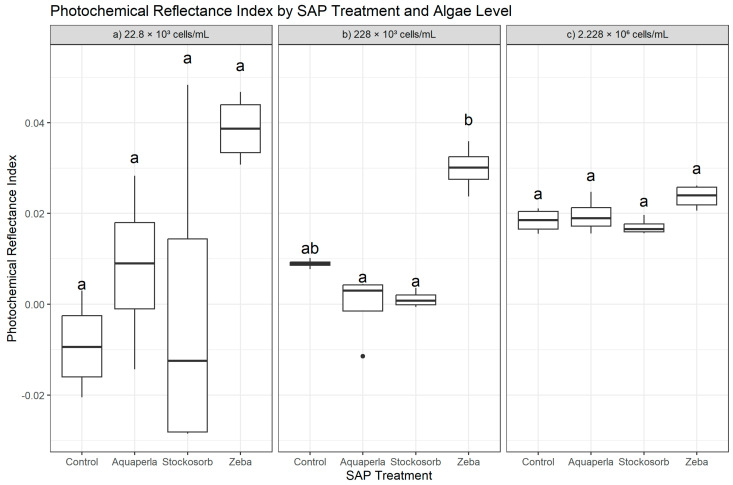
Cross-sectional Photochemical Reflectance Index (PRI) by SAP within algal densities (averaged over weeks 2–7). Boxes show median and interquartile range; whiskers extend to 1.5 × IQR; points are individual observations. Panels correspond to algal densities ((**a**): 22.8 × 10^3^; (**b**): 228 × 10^3^; (**c**): 2.228 × 10^6^ cells/mL). Letters above boxes denote groups not significantly different at α = 0.05 based on Dunn–Bonferroni pairwise tests following density-specific Kruskal–Wallis analyses.

**Figure 6 plants-14-02962-f006:**
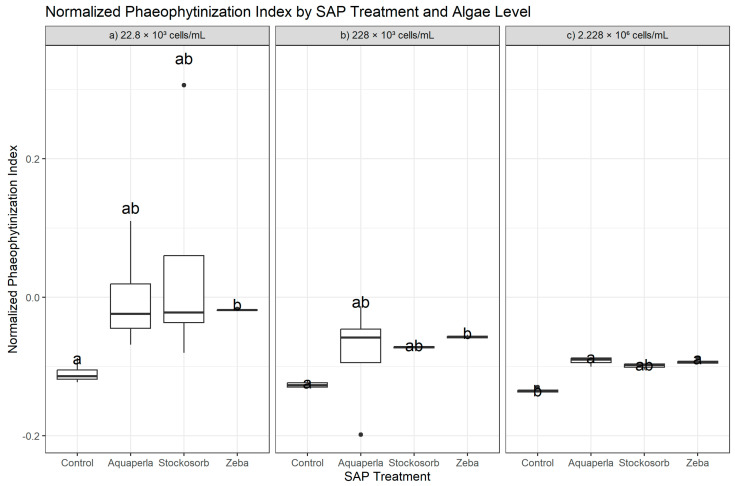
Cross-sectional NPQI by SAP within algal densities (averaged over weeks 2–7). Boxes show median and interquartile range; whiskers extend to 1.5 × IQR; points are individual observations. Panels correspond to algal densities ((**a**): 22.8 × 10^3^; (**b**): 228 × 10^3^; (**c**): 2.228 × 10^6^ cells/mL). Letters above boxes denote groups not significantly different at α = 0.05 by Dunn–Bonferroni pairwise tests following density-specific Kruskal–Wallis analyses.

**Figure 7 plants-14-02962-f007:**
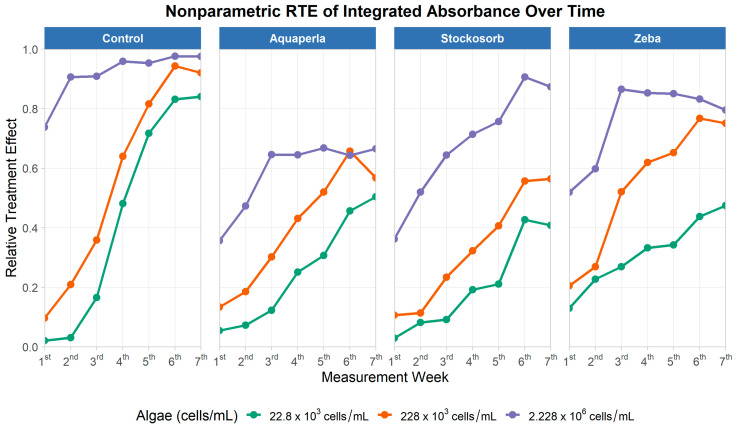
Relative treatment effects for integrated absorbance over seven measurement weeks, by SAP treatment and algae density. Panels show nonparametric RTE trajectories (0 = below, 1 = above the pooled median) for Control, Aquaperla, Stockosorb, and Zeba, with line colour indicating algae concentration (22.8 × 10^3^, 228 × 10^3^, 2.228 × 10^6^ cells/mL).

**Figure 8 plants-14-02962-f008:**
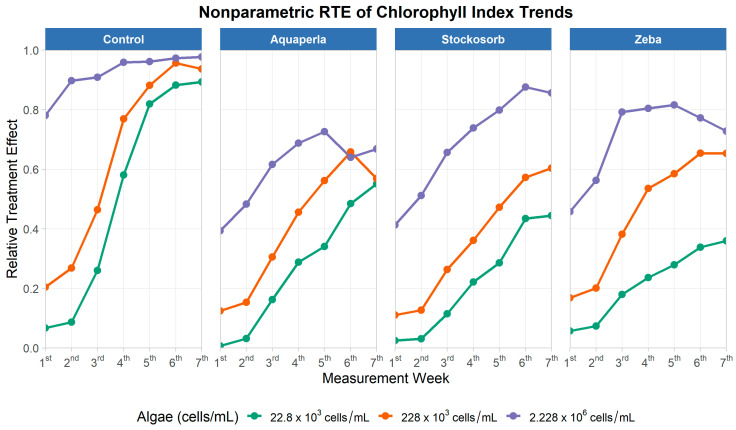
Relative treatment effects for chlorophyll index over seven measurement weeks, by SAP treatment and algae density. Panels show nonparametric RTE trajectories (0 = below, 1 = above the pooled median) for Control, Aquaperla, Stockosorb, and Zeba, with line color indicating algae concentration (22.8 × 10^3^, 228 × 10^3^, 2.228 × 10^6^ cells/mL).

**Figure 9 plants-14-02962-f009:**
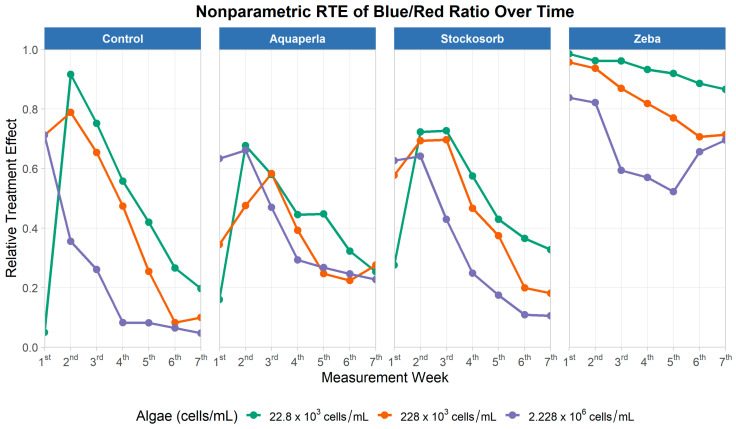
Relative treatment effects for blue/red absorbance ratio over seven measurement weeks, by SAP treatment and algae density. Panels show nonparametric RTE trajectories (0 = below, 1 = above the pooled median) for Control, Aquaperla, Stockosorb, and Zeba, with line color indicating algae concentration (22.8 × 10^3^, 228 × 10^3^, 2.228 × 10^6^ cells/mL).

**Figure 10 plants-14-02962-f010:**
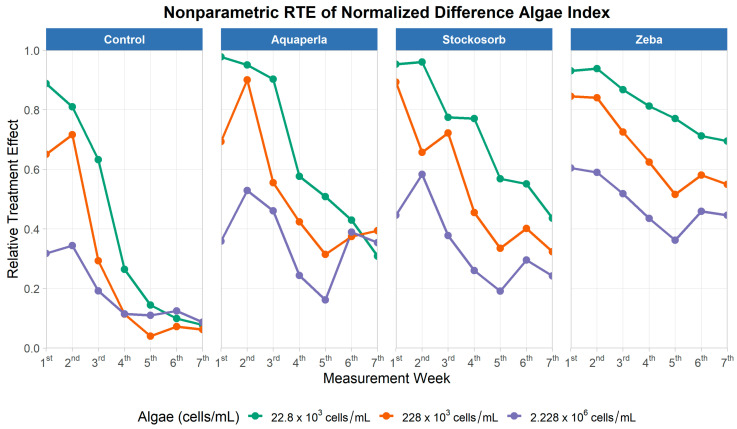
Relative treatment effects for normalized difference algal index over seven measurement weeks, by SAP treatment and algae density. Panels show nonparametric RTE trajectories (0 = below, 1 = above the pooled median) for Control, Aquaperla, Stockosorb, and Zeba, with line color indicating algae concentration (22.8 × 10^3^, 228 × 10^3^, 2.228 × 10^6^ cells/mL).

**Figure 11 plants-14-02962-f011:**
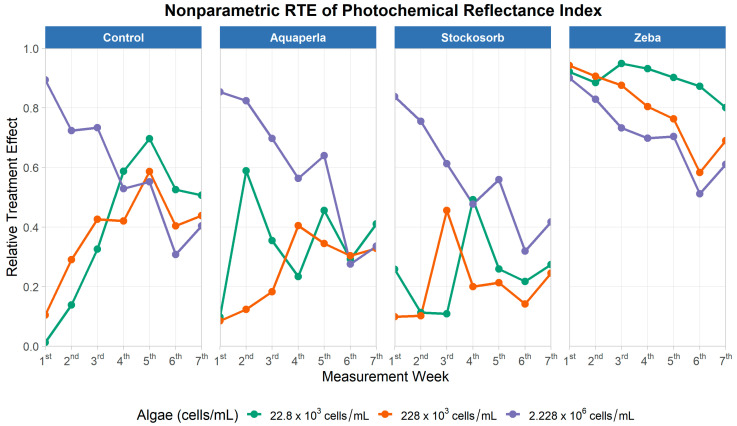
Weekly Relative Treatment Effects (RTEs) for the Photochemical Reflectance Index (PRI) over seven measurement weeks, shown by SAP treatment (panels: Control, Aquaperla, Stockosorb, Zeba) and algae density (lines: 22.8 × 10^3^, 228 × 10^3^, 2.228 × 10^6^ cells/mL). RTE (0–1) denotes the probability that observations from a group exceed the pooled distribution (0 = mostly below, 1 = mostly above the pooled median).

**Figure 12 plants-14-02962-f012:**
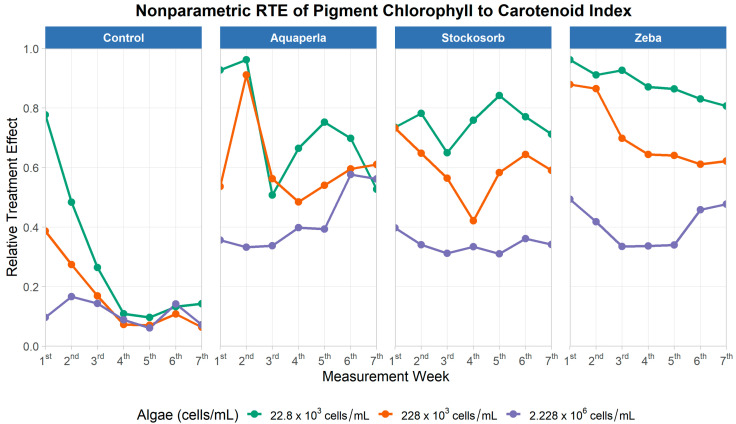
Weekly Relative Treatment Effects (RTEs) for the NPQI (chlorophyll-to-carotenoid index) across seven measurement weeks, stratified by SAP treatment (panels: Control, Aquaperla, Stockosorb, Zeba) and algae density (lines: 22.8 × 10^3^, 228 × 10^3^, 2.228 × 10^6^ cells/mL). RTE (0–1) is the probability that observations from a group exceed the pooled distribution (0 = mostly below, 1 = mostly above the pooled median).

**Figure 13 plants-14-02962-f013:**
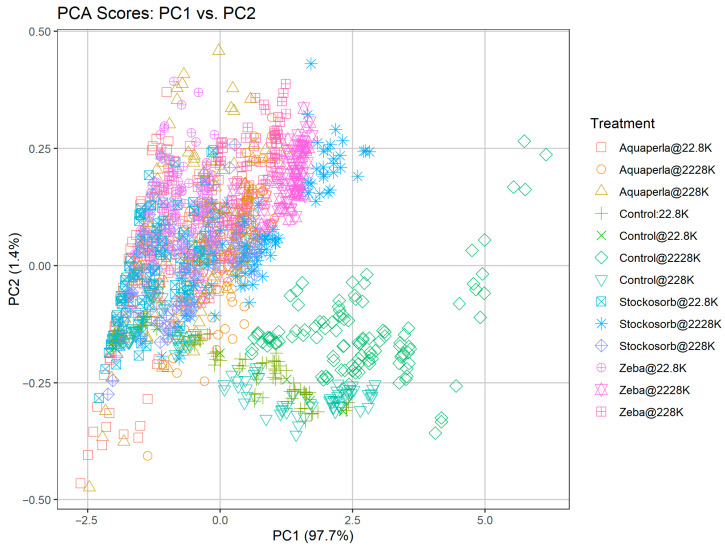
Principal component analysis score plot of raw absorbance spectra. Samples are plotted on PC1 (97.7% of variance) versus PC2 (1.4% of variance) axes, with each point colored and shaped according to its SAP treatment @ algal density. PC1 orders samples by biomass (lower densities on the left, higher on the right), while PC2 highlights subtler, treatment-specific spectral shifts.

**Figure 14 plants-14-02962-f014:**
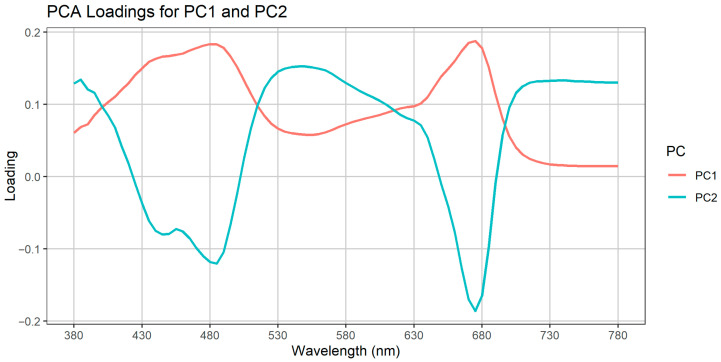
Loadings of the first two principal components from PCA of raw absorbance spectra (380–780 nm, 50 nm tick intervals). PC1 (red) exhibits strong positive loadings at chlorophyll-*a* absorption bands (~430–450 nm and ~670–680 nm), indicating that PC1 captures overall pigment concentration. PC2 (teal) displays negative loadings in the blue (≈450 nm) and red (≈680 nm) regions and positive loadings in the green–yellow region (≈525–600 nm), reflecting treatment-specific alterations in spectral shape.

**Figure 15 plants-14-02962-f015:**
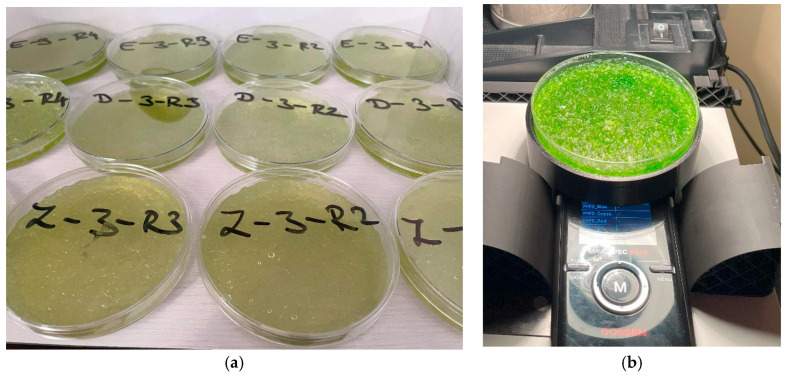
The mixed algae in the superabsorbent polymer (**a**); measurement of the spectral transmittance (**b**).

**Table 1 plants-14-02962-t001:** ANOVA-Type Statistic (ATS) results for the effects of SAP treatment, algae density, and time on integrated absorbance.

Effect	ATS	df	*p*
SAP	91.38	2.34	<0.001
Algae	456.76	1.81	<0.001
Time (Measurement)	761.83	2.95	<0.001
SAP × Algae	5.72	4.08	<0.001
SAP × Time	20.22	5.55	<0.001
Algae × Time	41.9	5.13	<0.001
SAP × Algae × Time	12.95	8.26	<0.001

Note. ATS = ANOVA-Type Statistic. *p*-values less than 0.001 are reported as *p* < 0.001.

**Table 2 plants-14-02962-t002:** Summary of Relative Treatment Effect (RTE) ranges for integrated absorbance across SAP treatments and algae densities.

Treatment	Algae Density	M	SD	RTE Range
Control	22.8 K	0.4	0.4	0.02–0.84
Control	228 K	0.6	0.4	0.10–0.94
Control	2228 K	0.9	0.1	0.74–0.98
Aquaperla	22.8 K	0.3	0.2	0.06–0.50
Aquaperla	228 K	0.4	0.2	0.13–0.66
Aquaperla	2228 K	0.6	0.1	0.36–0.67
Stockosorb	22.8 K	0.2	0.2	0.03–0.43
Stockosorb	228 K	0.3	0.2	0.11–0.56
Stockosorb	2228 K	0.7	0.2	0.36–0.91
Zeba	22.8 K	0.3	0.1	0.13–0.47
Zeba	228 K	0.5	0.2	0.21–0.77
Zeba	2228 K	0.8	0.1	0.52–0.87

Note. RTE = Relative Treatment Effect; M = mean; SD = standard deviation. Higher RTE values indicate a greater probability that observations from a treatment exceed the overall distribution. An RTE of 0.5 indicates no difference from the overall median; RTE values approaching 1.0 indicate a very high probability that the treatment group exceeds the overall distribution, and RTE values approaching 0.0 indicate a very low probability (the treatment group is generally lower than the overall distribution).

**Table 3 plants-14-02962-t003:** ANOVA-Type Statistic (ATS) results for the effects of SAP treatment, algae density, and time on the chlorophyll index.

Effect	ATS	df	*p*
SAP	310.3	2.87	<0.001
Algae	1149.76	1.88	<0.001
Time (Measurement)	1159.66	2.35	<0.001
SAP × Algae	11.03	4.93	<0.001
SAP × Time	21.57	4.43	<0.001
Algae × Time	62.35	4.18	<0.001
SAP × Algae × Time	17.19	6.63	<0.001

Note. ATS = ANOVA-Type Statistic. *p*-values less than 0.001 are reported as *p* < 0.001.

**Table 4 plants-14-02962-t004:** Summary of Relative Treatment Effect (RTE) for the chlorophyll index across SAP treatments and algae densities.

Treatment	Algae Density	M	SD	RTE Range
Control	22.8 K	0.5	0.4	0.07–0.89
Control	228 K	0.6	0.3	0.20–0.96
Control	2228 K	0.9	0.1	0.78–0.98
Aquaperla	22.8 K	0.3	0.2	0.01–0.55
Aquaperla	228 K	0.4	0.2	0.12–0.66
Aquaperla	2228 K	0.6	0.1	0.39–0.73
Stockosorb	22.8 K	0.2	0.2	0.03–0.44
Stockosorb	228 K	0.4	0.2	0.11–0.60
Stockosorb	2228 K	0.7	0.2	0.41–0.88
Zeba	22.8 K	0.2	0.1	0.06–0.36
Zeba	228 K	0.5	0.2	0.17–0.65
Zeba	2228 K	0.7	0.1	0.46–0.82

Note. RTE = Relative Treatment Effect; M = mean; SD = standard deviation. An RTE of 0.5 indicates no difference from the overall median; RTE values approaching 1.0 indicate a high probability that the treatment group exceeds the overall distribution, and RTE values approaching 0.0 indicate a low probability that the treatment group exceeds the overall distribution.

**Table 5 plants-14-02962-t005:** ANOVA-Type Statistic (ATS) results for the effects of SAP treatment, algae density, and time on the blue/red ratio.

Effect	ATS	df	*p*
SAP	195.56	2.48	<0.001
Algae	41.93	1.72	<0.001
Time (Measurement)	89.28	2.85	<0.001
SAP × Algae	6.2	3.49	<0.001
SAP × Time	6.07	4.47	<0.001
Algae × Time	19.94	4.26	<0.001
SAP × Algae × Time	2.99	6.06	0.006

Note. ATS = ANOVA-Type Statistic. *p*-values less than 0.001 are reported as *p* < 0.001.

**Table 6 plants-14-02962-t006:** Summary of Relative Treatment Effect (RTE) for the blue/red ratio across SAP treatments and algae densities.

Treatment	Algae Density	M	SD	RTE Range
Control	22.8 K	0.5	0.3	0.05–0.92
Control	228 K	0.4	0.3	0.08–0.79
Control	2228 K	0.2	0.2	0.05–0.71
Aquaperla	22.8 K	0.4	0.2	0.16–0.68
Aquaperla	228 K	0.4	0.1	0.22–0.58
Aquaperla	2228 K	0.4	0.2	0.23–0.66
Stockosorb	22.8 K	0.5	0.2	0.28–0.73
Stockosorb	228 K	0.5	0.2	0.18–0.70
Stockosorb	2228 K	0.3	0.2	0.11–0.64
Zeba	22.8 K	0.9	0	0.87–0.99
Zeba	228 K	0.8	0.1	0.71–0.96
Zeba	2228 K	0.7	0.1	0.52–0.84

Note. RTE = Relative Treatment Effect; M = mean; SD = standard deviation. An RTE of 0.5 indicates no difference from the overall median; RTE values approaching 1.0 indicate a high probability that the treatment group exceeds the overall distribution, and RTE values approaching 0.0 indicate a low probability that the treatment group exceeds the overall distribution.

**Table 7 plants-14-02962-t007:** ANOVA-Type Statistic (ATS) results for the effects of SAP treatment, algae density, and time on the normalized difference absorbance index.

Effect	ATS	df	*p*
SAP	47.23	1.58	<0.001
Algae	66.07	1.5	<0.001
Time (Measurement)	99.94	3.63	<0.001
SAP × Algae	0.67	2.28	0.53
SAP × Time	3.39	4.95	0.005
Algae × Time	7.03	5.25	<0.001
SAP × Algae × Time	1.64	6.58	0.124

Note. ATS = ANOVA-Type Statistic. *p*-values less than 0.001 are reported as *p* < 0.001.

**Table 8 plants-14-02962-t008:** Summary of Relative Treatment Effect (RTE) for the normalized difference absorbance index across SAP treatments and algae densities.

Treatment	Algae Density	M	SD	RTE Range
Control	22.8 K	0.4	0.4	0.08–0.89
Control	228 K	0.3	0.3	0.04–0.72
Control	2228 K	0.2	0.1	0.09–0.34
Aquaperla	22.8 K	0.7	0.3	0.31–0.98
Aquaperla	228 K	0.5	0.2	0.31–0.90
Aquaperla	2228 K	0.4	0.1	0.16–0.53
Stockosorb	22.8 K	0.7	0.2	0.44–0.96
Stockosorb	228 K	0.5	0.2	0.32–0.89
Stockosorb	2228 K	0.3	0.1	0.19–0.58
Zeba	22.8 K	0.8	0.1	0.69–0.94
Zeba	228 K	0.7	0.1	0.52–0.85
Zeba	2228 K	0.5	0.1	0.36–0.60

Note. RTE = Relative Treatment Effect; M = mean; SD = standard deviation. An RTE of 0.5 indicates no difference from the overall median; RTE values approaching 1.0 indicate a high probability that the treatment group exceeds the overall distribution, and RTE values approaching 0.0 indicate a low probability that the treatment group exceeds the overall distribution.

**Table 9 plants-14-02962-t009:** ANOVA-Type Statistic (ATS) results for the effects of SAP treatment, algae density, and time on the Photochemical Reflectance Index.

Effect	ATS	df	*p*
SAP	120.57	2.75	<0.001
Algae	43.24	1.69	<0.001
Time (Measurement)	9.53	4.29	<0.001
SAP × Algae	14.9	4.02	<0.001
SAP × Time	3.69	6	0.001
Algae × Time	18.51	5.59	<0.001
SAP × Algae × Time	3.84	6.67	0.000

Note. ATS = ANOVA-Type Statistic. *p*-values less than 0.001 are reported as *p* < 0.001.

**Table 10 plants-14-02962-t010:** Pairwise Wilcoxon rank-sum tests comparisons of absolute mean differences in average PRI across algae densities within each SAP treatment. Cells show the mean difference (row–column) and BH-adjusted significance (* *p* < 0.05; ns = not significant).

SAP	22.8 × 10^3^ vs. 228 × 10^3^	22.8 × 10^3^ vs. 228 × 10^3^	228 × 10^3^ vs. 2.228 × 10^6^
Control	0.001 ns	0.007 ns	0.005 ns
Aquaperla	0.079 ns	0.061 ns	0.018 ns
Stockosorb	0.003 ns	0.017 *	0.014 *
Zeba	0.011 ns	0.018 ns	0.007 ns

**Table 11 plants-14-02962-t011:** ANOVA-Type Statistic (ATS) results for the effects of SAP treatment, algae density, and time on the Pigment Chlorophyll-to-Carotenoid Index.

Effect	ATS	df	*p*
SAP	133.37	2.35	<0.001
Algae	113.14	1.76	<0.001
Time (Measurement)	13.21	3.09	<0.001
SAP × Algae	4.76	3.78	0.001
SAP × Time	2.37	6.48	0.023
Algae × Time	4.13	4.97	0.001
SAP × Algae × Time	1.89	8.7	0.05

Note. ATS = ANOVA-Type Statistic. *p*-values less than 0.001 are reported as *p* < 0.001.

**Table 12 plants-14-02962-t012:** Pairwise Wilcoxon rank-sum tests of absolute mean differences in average NPQI across algae densities within each SAP treatment. Cells show the mean difference (row–column) and BH-adjusted significance (* *p* < 0.05; ns = not significant).

SAP	22.8 × 10^3^ vs. 228 × 10^3^	22.8 × 10^3^ vs. 228 × 10^3^	228 × 10^3^ vs. 2.228 × 10^6^
Control	0.008 ns	0.012 ns	0.004 ns
Aquaperla	0.023 ns	0.070 *	0.048 *
Stockosorb	0.034 *	0.059 *	0.025 *
Zeba	0.037 *	0.072 *	0.035 *

**Table 13 plants-14-02962-t013:** Results of the Principal Component Analysis.

Component	Standard Deviation	Proportion of Variance (%)	Cumulative Variance (%)
PC1	1.416	97.70	97.70
PC2	0.167	1.36	99.06
PC3	0.115	0.65	99.71
PC4	0.055	0.15	99.86
PC5	0.031	0.05	99.91

**Table 14 plants-14-02962-t014:** Experimental setup of the *Chlorella vulgaris*–SAP growth study.

Treatment	SAP Type/Composition	Algal Density (Cells/mL)	pH	EC (ppm)	EC (mS cm^−1^)	Replicates	Volume per Replicate
Control	— (no SAP)	22.8 × 10^3^	7.0	391.75	0.7835	4	32 mL
Control	— (no SAP)	228 × 10^3^	7.0	386.25	0.7725	4	32 mL
Control	— (no SAP)	2.228 × 10^6^	8.0	389.25	0.7785	4	32 mL
Aquaperla	Potassium polyacrylate	22.8 × 10^3^	7.0	391.75	0.7835	4	32 mL
Aquaperla	Potassium polyacrylate	228 × 10^3^	7.0	386.25	0.7725	4	32 mL
Aquaperla	Potassium polyacrylate	2.228 × 10^6^	8.0	389.25	0.7785	4	32 mL
Zeba	Starch-g-poly(2-propenamide-co-2-propenoic acid) potassium salt	22.8 × 10^3^	7.0	391.75	0.7835	4	32 mL
Zeba	Starch-g-poly(2-propenamide-co-2-propenoic acid) potassium salt	228 × 10^3^	7.0	386.25	0.7725	4	32 mL
Zeba	Starch-g-poly(2-propenamide-co-2-propenoic acid) potassium salt	2.228 × 10^6^	8.0	389.25	0.7785	4	32 mL
Stockosorb	γ-polyglutamate	22.8 × 10^3^	7.0	391.75	0.7835	4	32 mL
Stockosorb	γ-polyglutamate	228 × 10^3^	7.0	386.25	0.7725	4	32 mL
Stockosorb	γ-polyglutamate	2.228 × 10^6^	8.0	389.25	0.7785	4	32 mL

## Data Availability

All data are available upon request to the corresponding author.

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
