# Peer review of "Effects of Superabsorbent Polymers on Growth and Pigment Allocation in Chlorella vulgaris"

_plants, 2025, doi:10.3390/plants14192962_

Round 1

Reviewer 1 Report

Comments and Suggestions for Authors

This manuscript presents three superabsorbent polymers (SAPs) in the application of the algae growth. The research is meaningful to water environment and may rise some aspiration to the plant cultivation due to the intense relationship between algae and plants. Unfortunately, the authors have not expressed well and clear to excavate the deep reasons to their important discovery. The experiment design is not reasonable, and the results is not commendable. More information and explanations should be complimented to improve the manuscript quality. Due to the respect and considering for their hard working in this work, I suggest a major revision before the manuscript to be accept in Plants. And the following issues should be addressed.

  1. The language should be polished to improve readability and fluency. Professional English editing services or a native English writer is demanded to improve the manuscript.
  2. The reason to choose these three SAPs is not clear. Although the authors have presented diverse advantages of these SAPs in the introduction part, for example hydrogel-immobilized microalgae for wastewater treatment or bioactive compound production with SAPs and C. vulgaris, I still can not understand the logic line the authors presented in this work about why they start this research. Any way the current investigation mainly on the aspect of immobilizing C. vulgaris on the SAPs for water treatment. Thus, the expression about the introduction needs to be deep reconsidered and reorganize it.
  3. The manuscript requires thorough proofreading for grammar, clarity, and consistency (e.g., “Evonik” vs. “Evonik Stockosorb”, “Zeba” vs. “Zeba Parkmaster”).
  4. All the discussions are more like an experiment report than a scientific paper. I can not find enough discussion on revealing the interaction between SAP and microbes.
  5. All the reference part needs to be checked thoroughly, the missed page number (eg. Ref 3), the unexpected all uppercase words (eg. Ref 6), the uneven Journal output (eg. Ref 7), even some non-SCI papers exist in the reference part (eg. Ref 5, Advances in Agriculture and Environmental Science: Open Access). Please revise all these errors and unsuitable citations.
  6.  The figures' expression must be reconsidered. Too large blank margin is existed in the description part.
Comments on the Quality of English Language

The language should be polished to improve readability and fluency. Professional English editing services or a native English writer is demanded to improve the manuscript. 

Author Response

Dear Professor, 

Enclosed please find our detailed answer for your kidn reveiw. 

Sincerely yours,
The Authors

Reviewer 2 Report

Comments and Suggestions for Authors

Dear Authors,

I was asked by the Editors of "Plants" to review your manuscript titled "Effects of Superabsorbent Polymers on Growth and Pigment Allocation in Chlorella vulgaris". The manuscript addresses an important question of the effect of superabsorbent polymers (SAP) on microorganisms, in this case, unicellular algae. I found the manuscript to be well-written, interesting, and of scientific importance, especially in the area of bioremediation and biotechnology in general. The statistics are particularly robust and convincing. However, I have some concerns which I will share with the authors point-by-point.

Major concerns

L76 - what are the major implications of changes in pigment allocation? How does this hypothesis tie to your research? Please, elaborate this. 

L80 - In my opinion, placing Results before Methodology like this is jarring. It makes it more difficult to follow your experimental design and ensuing results. I suggest a re-organization of the manuscript following the standard IMRaD pattern (introduction, methodology, results, and discussion) which, I believe, is the recommendation by the journal itself. Also, try to start every subsection in the Results with a short introductory explanation or a reminder as to why you analyzed these parameters, with what goal, etc. Moreover, I believe the PCA analysis should be placed at the end of the Results, as it mostly serves as a form of summary, a synthesis of multiple data points. However, this is merely a suggestion.

L275 - the authors use PRI and NPQI which were developed for vascular plants. Can the authors elaborate whether the unique morphology and physiology of Chlorella as unicellular algae affects these parameters? Are there similar studies which the authors can reference? Did the authors implement mathematical corrections to these measurements for Chlorella? If there are limitations, the authors should emphasize them. 

Discussion 

Discussion is very short, which is a pity since the authors presented a lot of results that can (and must) be analyzed and discussed. 

L598 - authors did not measure nutrient availability and hydrogel stability in this study. I would suggest the authors to tone down this conclusion, to emphasize this limitation in the main text, and to cite similar studies where nutrients and hydrogel stability were measured. At the moment, this hypothesis is highly speculative. Also, "without suppressing metabolic activity" - this can only be claimed if metabolomics and transcriptomics were performed. 

L611 - can the authors elaborate this further, by comparing the behavior of controls to test samples? Can the authors cite similar studies to strengthen their conclusions?

L620 - the authors should discuss in greater detail the biological meaning of the changes in Blue/Red Light ratio in this study.

L651 - the implications for biotechnological applications are very succinctly discussed. I would suggest expanding this a bit more. How do the observations from this study translate to potential practical applications?

L673 - were the SAPs sterile or not? If not, this should be briefly discussed as a potential confounding factor. Also, what was the light density during the experiment?

Minor concerns

Minor English language editing needed.

Mind the formatting of the references.

The "6. Patents" subsection should be erased.

Comments on the Quality of English Language

Minor English language editing needed.

Author Response

Dear Professor, 

Enclosed please find our detailed answer for your kind reveiw. We indicated our changes in the corrected manuscript. 

Sincerely yours,
The Authors

Reviewer 3 Report

Comments and Suggestions for Authors

This manuscript is a valuable contribution to the field, exploring the effects of superabsorbent polymers (SAPs) on microalgal growth, an area with significant implications for sustainable agriculture and biotechnology. The experimental design is sound, and the use of spectral indices and nonparametric statistical analysis is appropriate and well-executed. The results are clear and convincingly support the main hypothesis. I recommend the manuscript for publication with minor revisions to enhance clarity and polish the presentation.

Strengths:

Originality and Relevance: The study addresses a critical gap in knowledge by investigating SAP-microbe interactions in a non-immobilized system. The findings have direct relevance to integrated crop management and biofertilization, making the research timely and impactful.

Strong Methodology: The authors' use of a comprehensive approach, including a three-way factorial design, is commendable. The application of Principal Component Analysis (PCA) to spectral data is a particularly strong point, allowing for the clear separation of biomass effects (PC1, explaining >97% of variance) from more subtle, treatment-specific spectral shifts (PC2). This rigorous approach provides a solid foundation for the subsequent analysis of individual indices.

Clear and Well-supported Results: The results section is meticulously detailed, presenting clear evidence of SAP-specific effects. For instance, the consistent performance of Zeba, which matched or approached the control group's biomass, stands in contrast to DCM, which reduced biomass-related measures, especially at high densities. This selective influence of different polymer chemistries is a key finding.

Logical Structure: The paper's flow is excellent, guiding the reader from a well-articulated research question in the introduction to a robust methodology, followed by a clear presentation and discussion of the results. The conclusions effectively summarize the key findings and their broader implications.

Minor Revisions Suggested:

Clarity and Consistency: While the writing is strong overall, a final proofread is needed to correct minor grammatical errors and inconsistencies. For example, some sentences are grammatically incomplete or awkwardly phrased, such as the sentence on page 1 that reads, "...biodegradable and hybrid SAPs combining synthetic polymers with starch or chitosan balance high water-holding capacity with environmental". These issues should be corrected for professional polish.

Figure and Text Cohesion: The manuscript could benefit from more explicit connections between the text and the figures. For instance, while the PCA results are discussed in detail, the discussion would be even more impactful if the authors directly referenced specific clusters or trends visible in Figure 1 to support their narrative. For example, explicitly mentioning how DCM samples cluster distinctly from the control at high densities on the PC1 axis would strengthen the argument.

Detailed Methodological Information: The "Materials and Methods" section is informative, but it could be enhanced with additional details. While the paper mentions that the trial was conducted under controlled conditions of $24^{\circ}$C and a 16h light/8h dark photoperiod, a more precise description of the light source, its intensity, and its location relative to the cultures would be beneficial for replication.

Data Presentation: The results sections (e.g., Integrated Absorbance, Chlorophyll Index) are very detailed, including extensive statistical results from the Dunn-Bonferroni tests. While important, this level of detail can be visually overwhelming in the main text. Consider summarizing these statistical comparisons in a concise table in the supplementary materials to improve readability and allow the main text to focus on the key findings.

Author Response

(The authors gave the same response as above.)

Round 2

Reviewer 1 Report

Comments and Suggestions for Authors

This paper can be published directly. Good luck to the authors.

Author Response

Dear Professor, 

We sincerely thank you for your valuable time and constructive feedback during both rounds of the review process. We are very pleased to read that you now consider the paper suitable for publication.

In line with your suggestion regarding the English language, we have carefully revised the manuscript again to further improve clarity and readability. Minor grammatical and stylistic corrections were implemented to ensure smoother expression throughout the text.

We greatly appreciate your positive evaluation and supportive recommendation for publication.

Sincerely, 
The Authors

Reviewer 2 Report

Comments and Suggestions for Authors

Dear Authors,

I see that you significantly improved your manuscript. 

I only have some suggestions for minor editing of the English language.

Also, check the references once again for uniform formatting.

Comments on the Quality of English Language

Minor English language editing (minor language polishing).

Author Response

Dear Professor, 

We sincerely thank you for your careful second-round review and your supportive comments on the manuscript.

In response to your suggestions:

  • The English language was carefully polished with the help of a native English-speaking colleague, who corrected minor grammatical and stylistic issues throughout the text.

  • We also rechecked the references to ensure uniform formatting, and we followed the official MDPI Mendeley template for consistency.

We appreciate your positive evaluation and constructive remarks, which helped us to further improve the quality of our work.

Sincerely yours,
The Authors